# Estimating Enhancement Ratios of Nitrogen Dioxide, Carbon Monoxide and Carbon Dioxide using Satellite Observations

Cameron G. MacDonald[1,2,3], Jon-Paul Mastrogiacomo[1], Joshua L. Laughner[4], Jacob K. Hedelius[1,5], Ray Nassar[6], and Debra Wunch[1]

[1]University of Toronto, Department of Physics, Toronto, Ontario, Canada
[2]University of Waterloo, Department of Physics and Astronomy, Waterloo, Ontario, Canada
[3]Princeton University, Program in Atmospheric and Oceanic Sciences, Princeton, New Jersey, USA
[4]Jet Propulsion Laboratory, California Institute of Technology, Pasadena, California, USA
[5]Space Dynamics Laboratory, Utah State University, North Logan, Utah, USA
[6]Climate Research Division, Environment and Climate Change Canada, Toronto, Ontario, Canada

**Correspondence:** Debra Wunch (dwunch@atmosp.physics.utoronto.ca)

**Abstract.**

Using co-located space-based measurements of carbon dioxide ($CO_2$) from the Orbiting Carbon Observatory 2 and 3 (OCO-2/3) and carbon monoxide (CO) and nitrogen dioxide ($NO_2$) from the Tropospheric Monitoring Instrument (TROPOMI), we calculate total column enhancements for observations influenced by anthropogenic emissions from urban regions relative to clean background values. We apply this method to observations taken over or downwind of 27 large (> 1 million population) urban areas from around the world. Enhancement ratios between species are calculated and compared to emission ratios derived from four globally gridded anthropogenic emission inventories. We find that these global inventories underestimate CO emissions in many North American and European cities relative to our observed enhancement ratios, while smaller differences were found for $NO_2$ emissions. We further demonstrate that the calculation and intercomparison of enhancement ratios of multiple tracers can help to identify the underlying biases leading to disagreement between observations and inventories. Additionally, we use high-resolution $CO_2$ inventories for two cities (Los Angeles and Indianapolis) to estimate emissions of CO and $NO_2$ using our calculated enhancement ratios, and find good agreement with both a previous modelling study for the Los Angeles megacity and California Air Resource Board (CARB) inventory estimates.

## 1 Introduction

Improving air quality and reducing greenhouse gas emissions are focuses of environmental policy from global to municipal levels (Gurney et al., 2018a). Emissions inventories provide information about the distribution and sources of air pollution and greenhouse gas emissions as well as their trends over time. These inventories are constructed using bottom-up approaches: information on socio-economic activity is used alongside expected emissions factors for these activities to model emissions (Gurney et al., 2012; Janssens-Maenhout et al., 2019). Atmospheric measurements have been shown to be useful as part of top-down approaches in validating and refining these emissions inventories (McKain et al., 2012; Duren and Miller, 2012).

The expansion of the constellation of Earth-observing satellites taking measurements of greenhouse gases and air pollutants has led to observations over urban regions with unprecedented spatiotemporal coverage. Kort et al. (2012) used observations from the Greenhouse Gases Observing SATellite (GOSAT), launched January 2009, to measure enhancements of atmospheric carbon dioxide ($CO_2$) over megacities. Since the launch of the Orbiting Carbon Observatory-2 (OCO-2) in July 2014 (Crisp et al., 2004), further studies have characterized emissions from urban regions (e.g., Wu et al., 2018; Reuter et al., 2019). The Orbiting Carbon Observatory-3 (OCO-3) aboard the International Space Station (ISS) since May 2019 (Eldering et al., 2019) has provided additional observations of $CO_2$ in urban areas (Kiel et al., 2021). Satellite remote sensing of additional air pollutants has been greatly expanded with the launch of the TROPOspheric Monitoring Instrument (TROPOMI) aboard the Sentinel-5 Precursor (S5P) satellite on 13 October 2017 (Veefkind et al., 2012). Early investigations into the TROPOMI carbon monoxide (CO) and nitrogen dioxide ($NO_2$) products have shown the ability of TROPOMI to map concentrations of these air pollutants at the city-scale (e.g., Borsdorff et al., 2018; Zhao et al., 2020).

Enhancement ratios have been shown to be useful in evaluating the validity of emissions inventories and estimating emissions from a variety of anthropogenic sources of greenhouse gases and air pollutants. Wunch et al. (2009) used a ground-based remote sensing instrument to measure the diurnal variation of greenhouse gases within California's South Coast Air Basin (SoCAB) and calculate enhancement ratios between $CO_2$, CO, $CH_4$ and $N_2O$. Hedelius et al. (2018) used a combination of ground-based and satellite-based remote sensing instruments and a Lagrangian particle dispersion model to derive improved enhancement and emissions ratios between $CO_2$, CO and $CH_4$ for the SoCAB, while also demonstrating good agreement between ratios computed using different methods. Enhancement ratio methods involving both greenhouse gases (primarily $CO_2$) and air pollutants have also been used to investigate the combustion characteristics of anthropogenic activities; Silva and Arellano (2017) used satellite measurements of $CO_2$, $NO_2$ and CO to show a correlation between the dominant forms of combustion and both $NO_2$:CO and CO:$CO_2$ enhancement ratios in 14 regions from around the world. More recently, Lama et al. (2020) used measurements from TROPOMI to investigate burning efficiencies in six megacities by computing $NO_2$:CO enhancement ratios and comparing to emissions ratios from global inventories. Plant et al. (2022a) used TROPOMI methane and CO enhancements to assess methane emissions from several US cities.

In this paper, we describe a method to compute enhancement ratios between $CO_2$, CO and $NO_2$ over 27 large urban areas by combining measurements from three different space-based instruments. We use measurements of atmospheric $CO_2$ from OCO-2 and OCO-3 and measurements of CO and $NO_2$ from TROPOMI to measure anomalies over urban areas relative to a regional background. Results across multiple overpasses of these urban regions are used to derive enhancement ratios, which are then compared to ratios calculated from four global emissions inventories: the Emissions Database for Global Atmospheric Research (EDGAR), the Open Source Data Inventory for Anthropogenic $CO_2$ (ODIAC), the Fossil Fuel Data Assimilation System (FFDAS) and the Mapping Atmospheric Chemistry and Climate and CityZen (MACCity) inventory.

In section 2, we will describe the datasets and global emission inventories used in our analyses. In section 3, we will describe our approach to derive enhancement ratios between gases from satellite measurements. Section 4 will present the results of this analysis and section 5 will discuss the implications of these findings. Finally, section 6 will summarize our conclusions and suggest future work.

## 2 Data

### 2.1 OCO-2

We use measurements of the column averaged dry air mole fraction of carbon dioxide ($X_{CO_2}$) from OCO-2 (Crisp et al., 2004). OCO-2 was launched on 2 July 2014 into a sun-synchronous orbit with an equator crossing time of about 13:30 (ascending node) as part of the afternoon constellation (or A-train) of satellites. OCO-2 has been collecting science measurements of ($X_{CO_2}$) since 6 September 2014, collecting around 1 million total column observations per day (Crisp et al., 2017). The OCO-2 instrument includes three different grating spectrometers that measure reflected solar radiation in the near-infrared (NIR) and shortwave infrared (SWIR) spectral regions. The spectral bands include the $O_2$ A-Band from 0.7576—0.7726 $\mu$m, along with the "weak" and "strong" $CO_2$ bands measured at 1.5906—1.6218 and 2.0431—2.0834 $\mu$m, respectively. Measurements are taken in a horizontal row with 8 cross-track footprints, with 3 rows of observations collected every second. Each individual footprint has dimensions of approximately 2.25 km in the along-track direction and up to 1.29 km in the cross-track direction (depending on the satellite orientation). OCO-2 observes in three different modes of operation. In nadir mode, the observations are taken at the sub-satellite point and measurements taken over water are typically filtered out. In glint mode, the satellite makes observations near the point of the Earth's surface where sunlight is specularly reflected (Crisp et al., 2017; Eldering et al., 2019). Finally, OCO-2 can operate in a target mode, where a small area of the Earth is observed for several minutes while the satellite passes overhead. This mode is often used for the validation of measurements against ground-based remote sensing stations (e.g., Wunch et al., 2017).

For this study we use bias-corrected measurements of $X_{CO_2}$ from the OCO-2 Level 2 Lite files, version 9 (Kiel et al., 2019), accessed from the Goddard Earth Sciences Data and Information Services Center (GES-DISC) (https://disc.gsfc.nasa.gov/). The bias-correction process for OCO-2 adjusts $X_{CO_2}$ based on spurious correlations with retrieved aerosols, surface albedo and the difference between the vertical gradients of the retrieved and a priori $CO_2$ profiles. Version 9 includes an additional surface pressure based bias correction to account for pointing offsets which can cause greater uncertainties in regions with considerable topographic changes. Binary quality flags are provided in the files to indicate high and low accuracy measurements; for this study we only use measurements that have been flagged as "good".

### 2.2 OCO-3

We also use measurements of $X_{CO_2}$ from OCO-3 aboard the International Space Station (ISS). OCO-3 was launched to the ISS on 4 May 2019, and began providing science measurements on 6 August 2019. The OCO-3 instrument is a nearly identical spectrometer to OCO-2, measuring spectra in the $O_2$ A Band and weak and strong $CO_2$ bands to provide an eight footprint swath of parallelogram-shaped soundings measuring approximately 1.6 km in the cross-track direction by 2.2 km in the along-track direction, with 3 rows of observations taken every second (Eldering et al., 2019). OCO-2 pointing is carried out by maneuvers to the spacecraft, which is not possible on the ISS, thus OCO-3 is equipped with a pointing mirror assembly (PMA). In addition to the three observation modes described for OCO-2, the PMA enables OCO-3 to scan in an additional "Snapshot Area Map" (SAM) mode, where a two-dimensional area is swept out by adjacent swaths of measurements. This mode measures across

regions on the order of 100 km × 100 km, with the goal of capturing detailed maps of sources of $CO_2$ such as cities, fossil fuel
burning power plants and volcanoes. According to the OCO-3 SAM web-page (https://ocov3.jpl.nasa.gov/sams/index.php),
between 6 August 2019 and 30 June 2020 there were over 2000 SAM maneuvers executed by OCO-3, with around half of
these instances corresponding to sites influenced by anthropogenic sources of $CO_2$.

Bias-corrected measurements of $X_{CO_2}$ from the early (VEarly) release of the OCO-3 Version 10 Lite Files were accessed
from the NASA GES-DISC. We found that most instances of dense measurements over cities were obtained while the instru-
ment was operating in SAM mode (Eldering et al., 2019).

## 2.3   TROPOMI

In our analysis, we use measurements of $NO_2$ and CO retrieved from TROPOMI observations. TROPOMI was launched on
board the European Space Agency (ESA) Sentinel-5 Precursor (S5P) satellite on 13 October 2017 into a sun-synchronous orbit
with an equator crossing time of about 13:30 (ascending node) and has been providing science measurements since 30 April
2018. TROPOMI is a nadir-viewing grating spectrometer that measures Earth-reflected solar irradiance in three spectral bands:
In the ultra-violet and visible light (UV-Vis) band from 0.27–0.5 $\mu$m, an NIR band from 0.675–0.775 $\mu$m and a SWIR band
from 2.305–2.385 $\mu$m (Veefkind et al., 2012). Measurements in these bands enable quantification of CO and $NO_2$, as well
as methane ($CH_4$; Hu et al., 2016), sulfur dioxide ($SO_2$; Theys et al., 2017), formaldehyde (HCHO; De Smedt et al., 2018),
ozone ($O_3$; ESA, 2022), and additional aerosol properties. The 2600 km wide swath of TROPOMI allows for global coverage
every day (van Geffen et al., 2020) (before loss of data due to clouds).

### 2.3.1   TROPOMI CO

A subset of TROPOMI measurements from the SWIR band (2.315–2.338 $\mu$m) are used to infer the total column of CO, along
with corresponding column averaging kernels and error estimates under clear-sky conditions (Landgraf et al., 2016). From 30
April 2018 until 6 August 2019, TROPOMI CO pixels were 7 km × 7 km, with 215 cross-track pixels. From 6 August 2019
onward, the along-track resolution of the instrument was improved to 5.5 km. The TROPOMI data that we use is version 1 of
the data product.

TROPOMI total column CO values exhibit a stripe bias between adjacent rows of along-track observations (Borsdorff et al.,
2018). In addition to creating offsets to adjacent observations, the magnitude of the bias can change in the along-track direction.
To remove this bias we use the Fourier Filter De-striping (FFD) method described by Borsdorff et al. (2019) as a non-uniformity
correction. This algorithm involves taking the two-dimensional Fourier Transform (FT) of the CO measurements from a single
orbit, and filtering out modes with high frequency in the cross-track direction and low frequency in the along-track direction.
We identified a similar bias in the surface level values of the CO column averaging kernels, so FFD was applied to these values
as well.

The quality of the measurements of the total column of CO from TROPOMI are denoted by a quality assurance value
("qa_value") provided with each observation, with 0 indicating the lowest quality and 1 the highest. Following the product user
manual (https://sentinels.copernicus.eu/documents/247904/0/Sentinel-5P-Level-2-Product-User-Manual-Carbon-Monoxide/),

we use measurements with a qa_value of 0.7 or greater, which indicates clear sky conditions. Furthermore, we filter out measurements taken either entirely or partially over water. Finally, we convert the provided total column in mol/m$^2$ to total column dry-air mole fractions ($X_{CO}$) in ppb as described in Wunch et al. (2016) using the retrieved total column of water and the provided surface pressure for each observation.

### 2.3.2 TROPOMI NO$_2$

Observations over 0.405–0.465 $\mu$m from the UV-Vis spectrometer in TROPOMI are used to infer the tropospheric vertical column densities of NO$_2$ (van Geffen et al., 2020). TROPOMI NO$_2$ ground sampling distance is approximately 3.5 km in the cross-track, with the same along-track distance as the CO product. Similar to the TROPOMI CO product, the quality of measurements of the tropospheric column of NO$_2$ is described by a qa_value field. Following the NO$_2$ Product User Manual (https://sentinels.copernicus.eu/documents/247904/4682535/Sentinel-5P-Level-2-Product-User-Manual-Nitrogen-Dioxide/), we use only NO$_2$ measurements with a qa_value of 0.75 or greater. No stripe bias correction is needed for the NO$_2$ product, as corrections have already been applied to the values provided in the NO$_2$ Level 2 files. Again the column densities are converted to dry-air mole fractions using the retrieved column of water and reported surface pressure. Finally, tropospheric averaging kernels are derived from the provided total column averaging kernels following the method described by Eskes et al. (2019).

### 2.4 Cities

For information on the location and extent of cities from around the world, we use the European Commission Joint Research Centre's (EC-JRC) Global Human Settlement layer Urban Centres Database (GHS-UCDB) (Florczyk et al., 2019). Though "cities" in this database are often urban agglomerations composed of multiple municipalities, we refer to an urban agglomeration as a single city for convenience. Spatial extents of over 13000 cities are determined based on the presence and density of buildings and from the population density of the region from the GHS Built-up Areas (GHS-BUILT) and GHS population density (GHS-POP) databases (Corbane et al., 2018), respectively. Polygons defining the boundaries of each city are provided on a 1 km × 1 km grid. We focus our attention primarily on cities with total populations greater than 1 million persons, where we typically observe average $X_{CO_2}$ enhancements on the order of 1 ppm compared to nearby measurements that are not influenced by anthropogenic sources. We also look at lower population cities in North America and Europe which have high per capita emissions. While emissions for smaller cities are not investigated, their presence in the dataset is often useful in explaining additional enhancements or plumes observed within the data.

## 2.5 Emissions Inventories

Bottom-up emissions inventories are used as a separate way to derive emissions ratios. Emission inventories are typically on a mass basis, and need to be converted to a molar basis for comparison with satellite-derived estimates. We derive the inventory-based ratio A:B from

$$\alpha = \left( \frac{M_B}{M_A} \right) \frac{E_A^{\text{City, Inv}}}{E_B^{\text{City, Inv}}}, \tag{1}$$

where $M_A$ and $M_B$ are the molar masses of the respective species, and $E_A^{\text{City, Inv}}$ and $E_B^{\text{City, Inv}}$ are the total emissions estimates for the given city and inventory in mass per year. Total emissions for a city are derived by integrating the fluxes over the extent of the GHS polygon for the given city. We combine four different gridded global anthropogenic inventories to derive ratios for $NO_2$:CO, $NO_2$:$CO_2$ and CO:$CO_2$.

The Open Source Data Inventory for Anthropogenic $CO_2$ (ODIAC2018) is a high resolution global emissions inventory for $CO_2$ that provides anthropogenic fluxes on an approximately 1 km × 1 km grid with monthly temporal frequency from 2000–2019 (Oda and Maksyutov, 2011). Gridded fluxes are disaggregated from total country emissions using information on strong point sources and satellite observations of night lights. We further disaggregate the monthly estimates provided by ODIAC to weekly and diurnal time scales using the Temporal Improvements for Modeling Emissions by Scaling (TIMES) scaling factors (Nassar et al., 2013) which provide gridded (0.25°× 0.25°) scale factors based on both the day of the week and hour of the day.

The Fossil Fuel Data Assimilation System Version 2.2 (FFDAS) is a high resolution anthropogenic $CO_2$ emissions inventory which provides yearly fluxes for the period 1997–2015 on a 0.1°× 0.1° spatial grid (Asefi-Najafabady et al., 2014). Because the FFDAS time period does not extend to the operational period of TROPOMI, we use the 2015 values to derive our estimates. Similar to ODIAC, night lights data are used to disaggregate national emissions data down to a finer resolution. We apply the TIMES scaling factors to the disaggregated data in the development of FFDAS. Before being intergrated over the GHS polygon extent to get city-wide estimates, the FFDAS grid is downscaled to the ODIAC resolution, with uniform distribution across the original grid cell.

The EC-JRC Emissions Database for Global Atmospheric Research Version 5.0 (EDGARv5.0) is a global emissions inventory which provides gridded fluxes for many greenhouse gases and air pollutants for 1970–2015 (Crippa et al., 2020; European Commission and Joint Research Centre et al., 2019). We use the inventories for $CO_2$, CO and $NO_x$ (the combination of NO and $NO_2$). Emissions of $NO_2$ are approximated by dividing the provided $NO_x$ emissions by a factor of 1.32 (Pandis and Seinfeld, 2006). EDGAR is provided on a 0.1°× 0.1° grid with yearly fluxes for the time period 1970–2015. Similar to FFDAS, we apply TIMES scaling factors to the $CO_2$ inventory, and downscale the grid to the ODIAC resolution before integrating over a city region.

The final inventory that is used is the Mapping Atmospheric Chemistry and Climate and CityZen (MACCity, Granier et al., 2011). MACCity provides yearly fluxes of CO and $NO_x$ on a 0.5°× 0.5° global grid from 1990–2010. As with the other

inventories, MACCity was downscaled to the resolution of ODIAC before city emissions were derived. Table 1 summarizes emissions estimates from these inventories for all cities that are considered in this study.

## 3    Methods

### 3.1    Co-location of OCO-2 and TROPOMI Data

Because OCO-2 and S5P have sun-synchronous orbits with similar equator crossing times and repeat cycles, there is often
overlap between observations from the two instruments. We locate overpasses of cities by searching for OCO-2 observations within 75 km of a city boundary of interest. Winds at 50 metres are chosen to represent the boundary layer and are used to filter overpasses. We interpolate from the Modern-Era Retrospective analysis for Research and Applications Version 2 (MERRA-2, Molod et al., 2015) at a spatial resolution of $0.5°$ latitude $\times$ $0.625°$ longitude and 3-hourly temporal resolution to the location and time of the overpass. When the boundary layer wind direction does not intersect the OCO-2 ground track, the overpass
is rejected, as the pollution plume from the city will not be captured. Further filtering is performed to remove overpasses where the OCO-2 data downwind of the city is extremely sparse ( >95% of OCO-2 observations near the city flagged as bad), which is often the case when there is significant cloud cover in the region. After these filtering steps, the OCO-2 data, along with the TROPOMI data from the same time period are visualized and inspected to check for issues such as the presence of secondary sources of greenhouse gases or pollutants, and to ensure there are spatially coincident measurements between the
two satellites. Secondary sources from cities are identified using the European Commission Joint Research Centre's (EC-JRC) Global Human Settlement layer Urban Centres Database (GHS-UCDB) (Corbane et al., 2018). Secondary sources from power plants are identified using the Carbon Monitoring for Action (CARMA) database (Ummel, 2012). During this step, corrections are applied to the MERRA-2 wind direction if considerable discrepancies are observed between the given wind bearing and the behaviour of the plume emanating from the city, which is generally most visible in the TROPOMI $NO_2$ product. Similar
manual corrections have been employed in past studies using observations from OCO-2 (e.g., Nassar et al., 2017; Reuter et al., 2019; Nassar et al., 2021); these errors in wind direction can be caused by the inability of the coarse model resolution to resolve local topography, or if the 50-m winds are not representative of the winds at the local plume height. The wind rotation we perform should at least partially correct for both these errors. To compute enhancement ratios, coincident TROPOMI CO and $NO_2$ enhancements are selected at the locations of the OCO-2 ground track (Figure 1). At best, our manual inspections
found that 83% of the overpasses which passed the initial automatic filtering were viable in a city, and at worst, all of the overpasses were rejected for a city. The median retention rate is 31% of the overpasses per city.

A similar approach is used to search for co-located measurements from OCO-3 and TROPOMI. This task is more complex as the ISS is in a different type of orbit than OCO-2 and S5P, and thus does not consistently take co-located measurements with the instruments in sun-synchronous orbits. This can lead to much greater time differences between co-located OCO-3 and
TROPOMI measurements compared to the differences between observations from OCO-2 and TROPOMI. Upon identifying a favorable OCO-3 overpass map of a city, the TROPOMI track which lies closest to the city is selected. Time offsets are as large as 6 hours between observations from the two instruments. This leads to greater uncertainties in cases where the wind

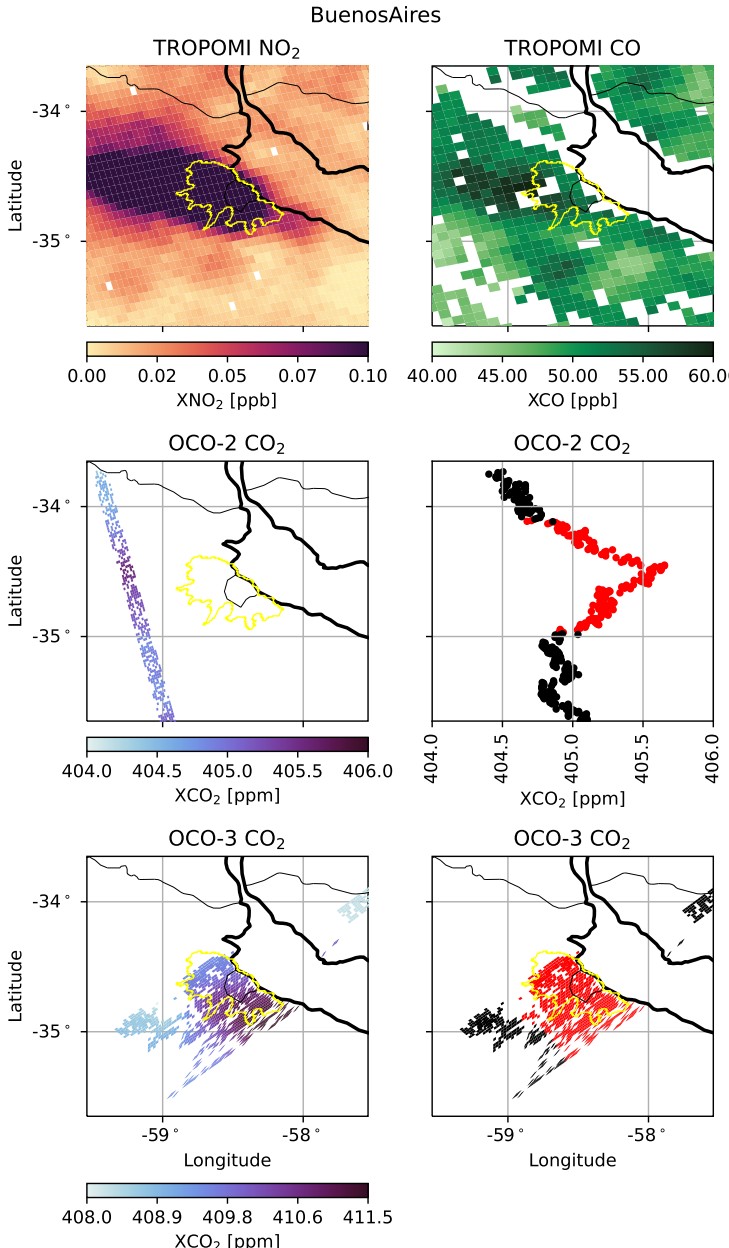

**Figure 1.** Examples of the various satellite instrument ground tracks over Buenos Aires taken on 11 January 2019 (top 2 rows) and 19 September 2019 (bottom row). The yellow outline indicates the city boundaries. The top two panels show the TROPOMI $NO_2$ (left) and $CO$ (right). The middle two panels show the OCO-2 ground track downwind of Buenos Aires (left), and the $CO_2$ measurements as a function of latitude (right). Red points indicate the urban enhancement plume and black points are considered outside the plume. The bottom two panels show the OCO-3 snapshot area mapping (SAM) mode measurements over Buenos Aires (left). The right panel distinguishes the region of enhanced $CO_2$ in red from the background region in black.

direction has changed significantly in the time between overpasses, as the regions that are affected by the city's plume may no longer coincide with one another. Coincident TROPOMI enhancements are selected at the locations of the OCO-3 SAM
measurements (Figure 1).

Finding instances of co-located measurements of $NO_2$ and CO from TROPOMI is a far simpler task. Here we search for instances of measurements directly over each city of interest, leading to 1 or sometimes even 2 overpasses per day, depending on the longitude of the ground tracks. Due to this much higher data volume, only direct observation of the cities within their bounding areas are considered when deriving $NO_2$:CO enhancement ratios to avoid including measurements which are not
influenced by the cities due to wind direction errors and lifetime effects. Figure 1 provides example measurement footprints from OCO-2, OCO-3, and S5P's TROPOMI instrument.

### 3.2   Identifying Enhancements

To identify the subset of measurements which are influenced by emissions from the city, we transform the latitude-longitude positions of the observations into along- and cross-wind distances from the city centre and use the equation for the spread of a
vertically-integrated Gaussian plume, defined by Krings et al. (2011) as

$$\sigma_y = a\left(\frac{x + x_0}{x_c}\right)^{0.894}, \tag{2}$$

where $x$ is the downwind distance in metres, $x_c = 1000$ m is a characteristic length scale and $a$ is the atmospheric stability parameter (Pasquill, 1961; Krings et al., 2011; Nassar et al., 2017, 2021) which controls the spread of the plume based on the observed MERRA-2 50 m wind speed, so that the plume changes width depending on the wind speed and insolation.
Following Nassar et al. (2017), we use the Pasquill-Gifford stability class to determine the atmospheric stability parameter (Martin, 1976), and as in Nassar et al. (2021) we assume solar insolation to be strong given the clear-sky requirements for dense OCO-2/3 observation. The distance $x_0$ is used to define the initial width of the plume and is defined by

$$x_0 = x_c\left(\frac{y_0}{4a}\right)^{\frac{1}{0.894}}, \tag{3}$$

where $y_0$ is the cross-wind extent of the city in metres. The factor of 4 in the definition of $x_0$ follows the method of Krings et al.
(2011) so that the cross-wind extent of the city is associated with a $\pm 2\sigma_y(x_0)$ spread of the plume. Downwind observations with cross-wind distances that are less than $2\sigma_y(x)$ from the mean path from the city $y$ are considered to be in the plume. In cases where the winds point parallel to the affected OCO-2 track, a maximum downwind distance for the plume is determined manually, following Nassar et al. (2017, 2021) to visually identify a drop in $XCO_2$, which limits the length of the plume to an area where significant enhancements are observed.
For comparisons between the TROPOMI $NO_2$ and CO products, where we do not consider a plume region, the enhancement area is taken as the bounding box of the GHS polygon for the city.

### 3.3 Anomaly Calculation

#### 3.3.1 Smoothing of Urban Influenced Data

To decrease the amount of noise in the OCO-2/3 and TROPOMI data, we apply a nearest neighbour fit with a constant radius to smooth out the data (Altman, 1992). For the narrow swath width of OCO-2, we find that fitting a surface to the OCO-2 time series and using a radius of 2 seconds (equivalent to about 6 rows of OCO-2 measurements) leads to a fit that removes high frequency noise but retains the overall trends (Figure 2). Due to the wide swath width of TROPOMI, it is more appropriate to fit a spatial surface to the data. We find that using a radius of 15 km effectively smooths the data, and is comparable in spatial extent to the smoothing applied to the OCO-2 data. Fitting these surfaces to the datasets has the added advantage that predictions can be made at locations which do not have direct measurements, but have adequate nearby coverage. This lessens the impact of spurious missing data when trying to find co-located measurements.

#### 3.3.2 Background Calculation

A regional background for $X_{CO_2}$ is determined using nearby measurements from OCO-2 that are free from anthropogenic influence. A nearest neighbour fit similar to that used to smooth the urban influenced data is applied here, but with a much larger radius of 20 seconds (approximately 140 km). This choice of radius creates a background whose extent is similar to those in the simulations performed by Wu et al. (2018). The background fit is performed using typically the lowest 75 percent of the retrievals, so that potential enhancements are removed. For some individual overpasses, the choice of percent must be tuned if the nearest neighbor background fit gives a trend that appears to be too high or too low compared to nearby observations which are unaffected by any urban plumes. The performance of this method appears to be most accurate when winds run perpendicular to the OCO-2 track and there exist dense soundings both prior to and after crossing the plume of the city, however satisfactory results were still found in cases where soundings were missing on one side of the plume, which is often the case when cities close to a body of water are observed in nadir mode. Figure 2 shows an example of the process of smoothing the data, identifying an enhancement, and calculating the background for an overpass of Moscow with OCO-2.

A similar method is used to define the background for TROPOMI. A radius of 150 km is used for this fit, again using only the lowest 75 percent of data to avoid the influence of anthropogenic enhancements on the background. Due to the larger swath width and generally more dense measurements of the TROPOMI products, we find the value does not need to be tuned from 75 percent for individual overpasses.

#### 3.3.3 Calculation of Anomalies

Anomalies are calculated by subtracting the background estimates from the smoothed urban influenced values. Coincident observation locations are then chosen using the locations from the sparser of the two species being investigated. For example, if a ratio between CO and $CO_2$ is being determined, the observation locations from OCO-2 or OCO-3 are used, as their spatial coverage is much smaller than that of the TROPOMI CO product (Figure 1).

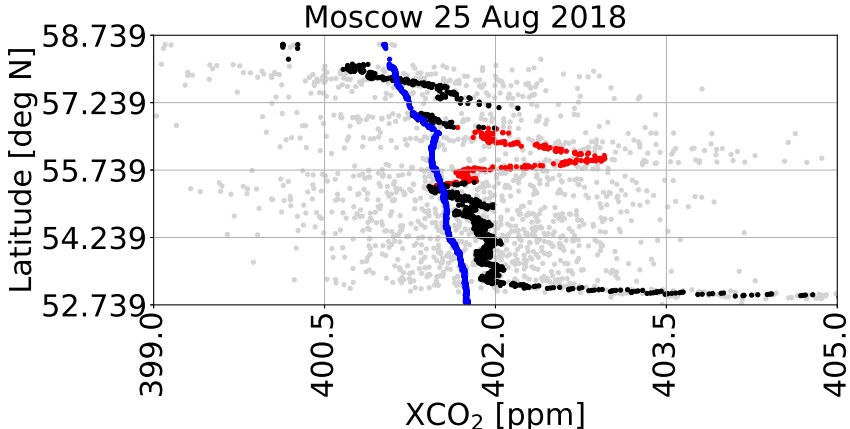

**Figure 2.** An example of of the background and anomaly calculation procedure for OCO-2 data over Moscow (Russia) on 25 August 2018. The grey points represent the original measurements from the OCO-2 Lite files. The OCO-2 ground track in this example is downwind of the city and perpendicular to the wind direction. The black and red points indicate the smoothed data, with the red points corresponding to where the Gaussian plume intersects the OCO-2 track. We consider the red points to be within the urban enhancement plume of the city. The blue line shows the derived background using the lowest 75 percent of data.

The $CO_2$ anomalies are divided by the column averaging kernel values at the surface pressure of the measurement, which is similar to the method used by Wunch et al. (2009) to account for the sensitivities of the instruments to changes in trace gas concentrations near the surface of the Earth, where the emissions from cities originate. We must also account for the fact that the TROPOMI a priori profiles of $NO_2$ and CO are extracted from the TM5 chemical transport model and thus contain spatial information such as urban enhancements. This requires that an additional correction term is added when inferring the true enhancement, $\Delta c^t$, from the retrieved enhancement, $\Delta \hat{c}$, if the urban and background a priori total columns, $c_u^a$ and $c_b^a$, and the surface pressure column averaging kernels, $a^0$, are known:

$$\Delta c^t = \frac{\Delta \hat{c}}{a^0} - \frac{(1 - a^0)(c_u^a - c_b^a)}{a^0} \tag{4}$$

Figure 3 demonstrates this anomaly calculation procedure, and Figure 4 shows an example of the distributions of surface averaging kernels for the three gases that are considered. Appendix §C describes the averaging kernel correction in detail.

### 3.4 Determination of Enhancement Ratios

To determine enhancement ratios, we aggregate all overpasses for a given city and regress one set of anomalies onto the other using a reduced major axis regression as described by York et al. (2004), as shown in Figure 5. The variance of the samples is used as the uncertainty for each observation. Reduced major axis regression has the property that the resulting slope is independent of which variable is chosen to be on the abscissa and which is chosen to be on the ordinate axis. Furthermore, the calculated slope is unaffected by the scaling of axes by a constant value, so that the calculated slopes are independent of the choice of mixing ratio units that are used (i.e., ppm or ppb). This method is then bootstrapped (Efron and Gong, 1983)

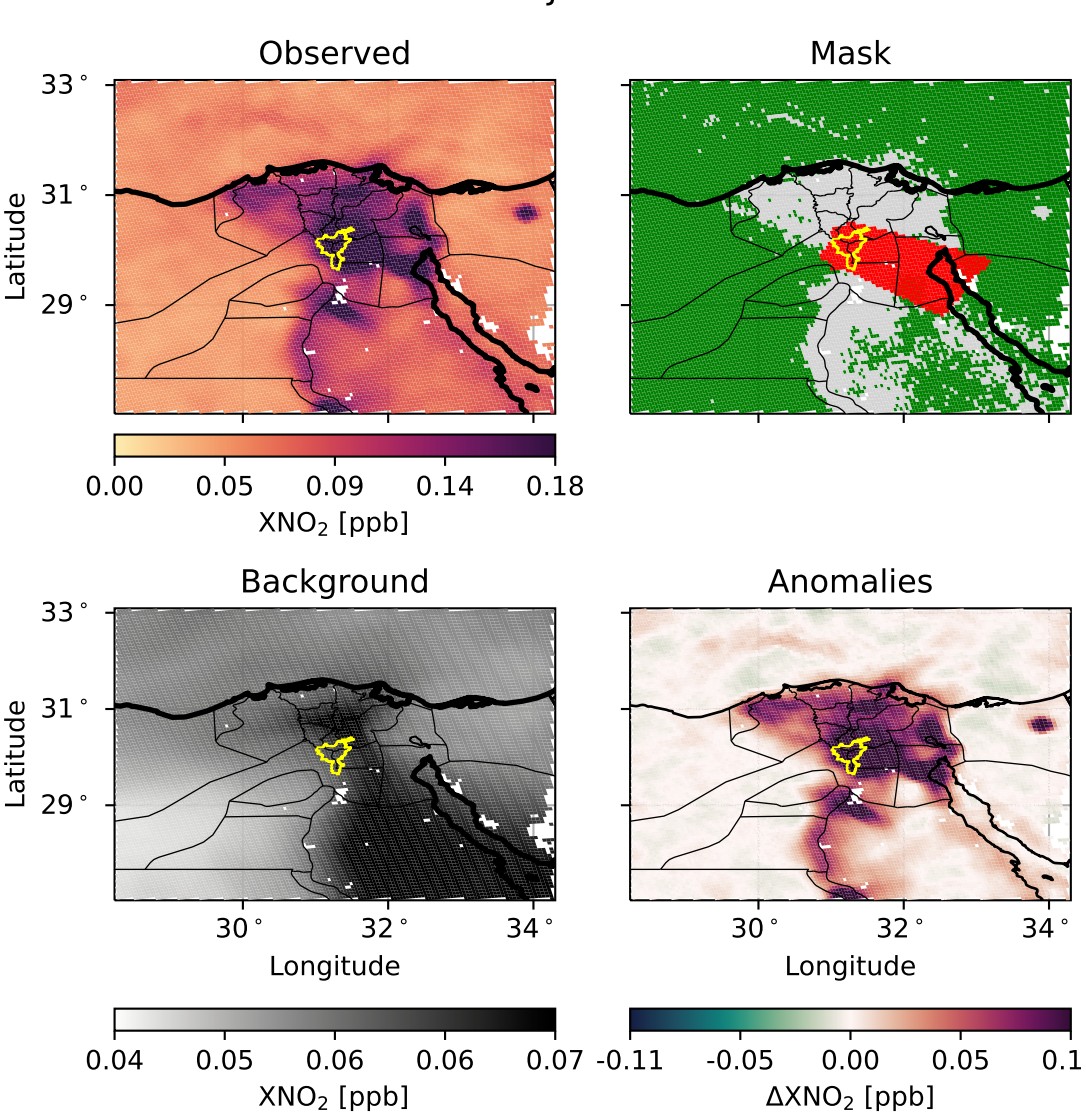

**Figure 3.** An example of the background and anomaly calculation procedure for TROPOMI NO$_2$ observations over Cairo on 15 June 2019. The top left panel shows the smoothed set of observations. The top right shows the mask for the background calculation; green points are below the $75^{th}$ percentile and used in the calculation while grey points are omitted, and red points indicate observations in the enhancement. The selection of the enhancement region is done liberally, as the location of OCO-2/3 observations is often a more limiting factor. The lower left panel then shows the derived background, and the lower right panel the calculated anomalies. The extent of the GHS polygon for Cairo is shown in yellow. Thick black lines mark coastlines, and thin black lines indicate geopolitical boundaries.

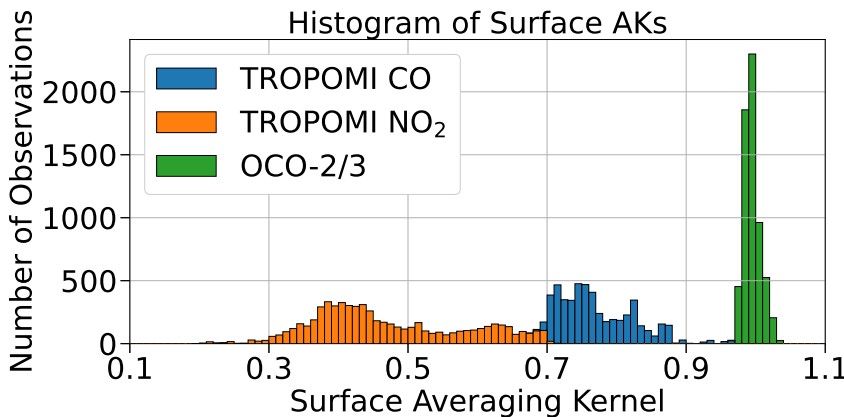

**Figure 4.** Example histograms of the averaging kernel values at the surface pressure for OCO-2/3 and TROPOMI from all the data collected over Phoenix (USA), which are used to account for the sensitivity of the instruments to concentration changes at the surface.

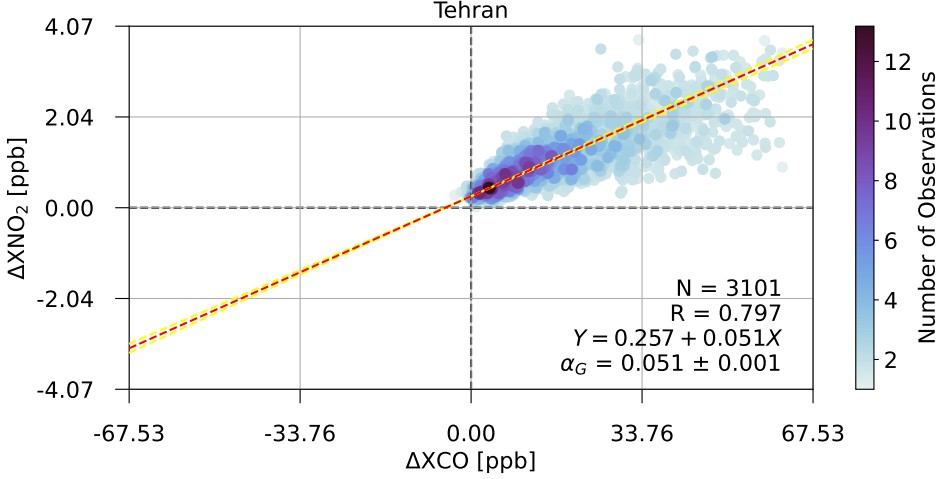

**Figure 5.** Regression of anomalies of $X_{NO_2}$ onto anomalies of $X_{CO}$ over Tehran using TROPOMI data from June–August 2018. The red dashed line indicates the estimated enhancement ratio, while the yellow lines represent $\pm 2\sigma$ uncertainties in the ratio. Each data point represents a single observation, and is assigned a color to indicate the density of points by binning the observations into 100 bins across each axis, and counting the number of points in the bin in which the observation resides. N is the total number of points included in the regression, R is the correlation coefficient, Y is the equation of best fit, and $\alpha_G$ is the slope and $2\sigma$ uncertainty determined by bootstrapping.

500 times to get an error estimate for the fit. Bootstrapping is a re-sampling technique in which random pairs of anomalies are drawn with replacement and fit independently, and has been used in previous enhancement ratio studies (e.g., Wunch et al., 2009, 2016; Lama et al., 2020). We take twice the standard deviation of the resulting set of slopes as the uncertainty estimate for the fit.

## 3.5 $NO_2$ Lifetime Correction

$NO_2$ has a short atmospheric lifetime compared to those of both $CO_2$ and CO, and can be on the same order as the advective time scales associated with emissions on the scale of large cities. To account for this, we apply a correction to the observed enhancement ratios to model the effect of $NO_2$ lifetime. Following Lutsch et al. (2020), the multiplicative correction takes the form

$$C = \exp\left(\tau_A/\tau_{NO_2}\right), \tag{5}$$

where $\tau_A$ is a time scale for advection and $\tau_{NO_2}$ is the lifetime of $NO_2$. This uses the fact that the chemical loss of $NO_2$ can be modeled as $NO_{2,\text{orig}} \cdot \exp(-t/\tau_{NO_2})$. Thus, when $t = \tau_A$, the ratio of $NO_{2,\text{orig}}/NO_{2,\text{downwind}} = C$. We use the method described in Laughner and Cohen (2019) to calculate $NO_2$ lifetimes for each city separately for summer and winter (details given in Appendix §B). We apply a single lifetime correction by scaling the observed enhancement ratio so that $\alpha_{\text{Corrected}} = C\alpha$, with an advection time scale given by

$$\tau_A = \frac{1}{\overline{U}}\left(\frac{A}{\pi}\right)^{1/2}, \tag{6}$$

where $\overline{U}$ is the average wind speed averaged across all overpasses of the city and weighted by the number of observations in each overpass, and $A$ is the area of the city provided in GHS-UCDB so that the relevant length scale is the radius of the city if it were a perfect circle with area $A$. In applying these corrections, we have neglected the lifetimes of CO and $CO_2$, which are on the order of months and centuries, respectively, and therefore will have a negligible effect on the observed enhancement ratios. To account for errors in the $NO_2$ lifetimes as well as the wind speeds used to calculate the advective time scales, we add an additional 20% in quadrature to the initial enhancement ratio uncertainty when the lifetime correction is applied.

## 4 Results

### 4.1 $NO_2$:$CO_2$ and CO:$CO_2$ Ratios

Using these methods, we are able to quantify $NO_2$:$CO_2$ ratios from 22 cities, and CO:$CO_2$ ratios from 21 cities using co-located observations from TROPOMI, OCO-2 and OCO-3. A total of 174 overpasses occurring from April 2018 to May 2020 are used to derive $NO_2$:$CO_2$ ratios, with 140 consisting of observations from TROPOMI and OCO-2, and the remaining 34 involving TROPOMI and OCO-3. The most overpasses for an individual city are found for Phoenix (United States), where we found 20 usable overpasses; Toronto has the fewest, where only a single overpass is used. In many cases, overpasses involving OCO-3 had to be filtered out due to observed non-linear relations between the derived $X_{CO_2}$ anomalies and those from the two TROPOMI products for a given day. This issue is most likely caused by the greater time differences between overpasses of the ISS and S5P—the location and distribution of the plumes could change significantly between the two times at which measurements are made so that the measurements are no longer approximately co-located in time and space.

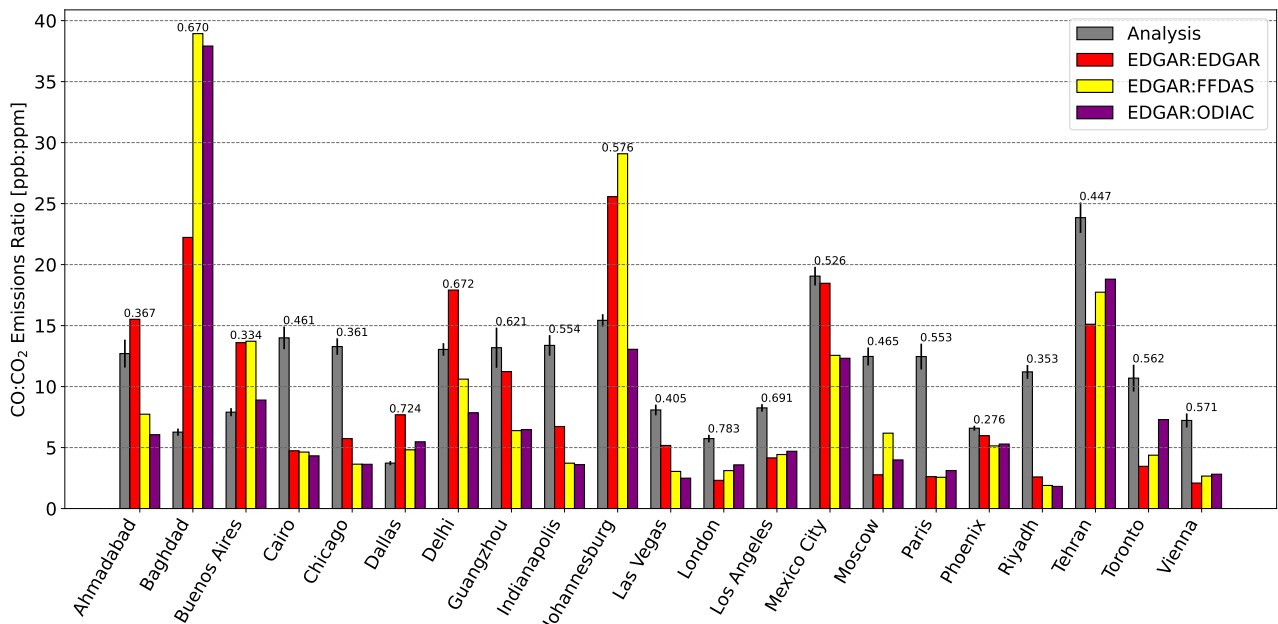

**Figure 6.** $CO:CO_2$ enhancement ratios derived using observations from OCO-2/3 and TROPOMI compared to inventory estimates using CO emissions from EDGAR and $CO_2$ emissions from EDGAR, FFDAS and ODIAC. The numbers on top of the bars denote the correlation coefficient between the two sets of observed anomalies. The error bars atop the bars represent the uncertainty in the measured ratios.

Cities located in the Southwestern United States (Los Angeles, San Francisco, Phoenix, and Las Vegas) and in the Middle East (Tehran, Baghdad, and Cairo), generally yielded more usable overpasses than cities located elsewhere in the world due to the greater number of cloud-free daylight hours in these regions. Few good overpasses were found for cities located in East and Southeast Asia, as observations from OCO-2/3 are often very sparse due to persistent clouds; other than Guangzhou (China) and Seoul (South Korea), megacities in these regions of the globe are not considered in this study. Overpasses were also generally better for inland cities compared to those situated next to large bodies of water, where loss of data from nadir viewing in the OCO-2/3 product and data filtering the TROPOMI CO product led to more limited opportunities to make observations downwind of the city. The presence of mountains near cities also presented some problems; in such cases large corrections to the reported MERRA-2 wind directions were often required to properly capture the plume. In the unique case of Los Angeles and the SoCAB in which the city resides, we considered only measurements directly above the SoCAB in our analysis, as the surrounding mountain ranges prevent air in the urban boundary layer from easily leaving the basin.

Figure 6 shows the derived $CO:CO_2$ enhancement ratios from our analysis compared to inventory based estimates of the enhancement ratios derived from the ODIAC, FFDAS and EDGAR emissions inventories. For cities across the United States, Canada, and Europe, we find that apart from the Dallas-Fort Worth area, our measured enhancement ratios are higher than the inventory-based estimates, with our results ranging from around 1.1 (Phoenix) to 4.7 (Paris) times greater than the respective EDGAR estimates. Results for the remainder of the cities show a less consistent picture, with the largest underestimate being

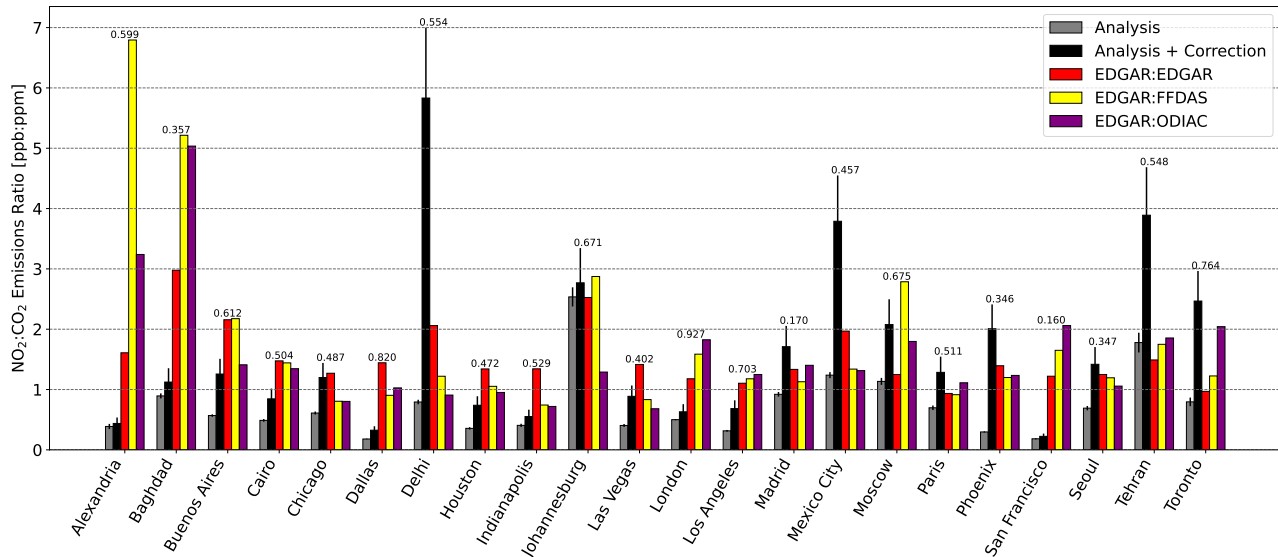

**Figure 7.** Same as in Figure 6 but for $NO_2$:$CO_2$ ratios derived using OCO-2/3 and TROPOMI observations. $NO_2$ enhancements are calculated without a lifetime correction in grey and labeled "Analysis," and with a lifetime correction in black labeled "Analysis + Correction."

for Baghdad (Iraq), where we find an enhancement ratio that is around 3 times lower than that of the EDGAR estimate, and
the largest overestimate for Tehran, where the measured enhancement ratio is around 1.6 times greater than the EDGAR value. When comparing to ratios calculated using CO emissions from the MACCity inventory, shown in Appendix §A, we find a more pronounced underestimation compared to observations across North America and Europe.

Figure 7 similarly shows results for the derived $NO_2$:$CO_2$ ratios, both with the correction for $NO_2$ lifetime and without. Here we find that without any $NO_2$ lifetime correction, almost all of the cities considered have derived ratios that are smaller than the
inventory-based estimates. As with the CO:$CO_2$ results, this discrepancy is most pronounced in the United States and Europe. Upon applying our lifetime correction to the ratios, the ratios in many of these cities are brought closer to the inventory-based estimates, with a few cities (Delhi, Mexico City and Tehran) having ratios that are higher than any of the inventory estimates. Delhi has a particularly low wind speed (Table 3), which may cause the $NO_2$ lifetime correction to be overestimated. Additionally, correlation coefficients between the two sets of anomalies were generally found to be greater in this case when
compared to the CO:$CO_2$ enhancement ratios results. Ratios calculated using the MACCity inventory, shown in Appendix §A, show better agreement with those derived using satellite observations prior to the application of the lifetime correction, but observed enhancements are generally higher than the inventory estimates after the atmospheric lifetime correction is applied.

## 4.2 $NO_2$:CO Ratios

We also derive $NO_2$:CO ratios for the same cities using only measurements from TROPOMI. Because $NO_2$:CO enhancement
ratios do not use observations from OCO-2/3 which have more limited coverage, there are far more opportunities for calculating

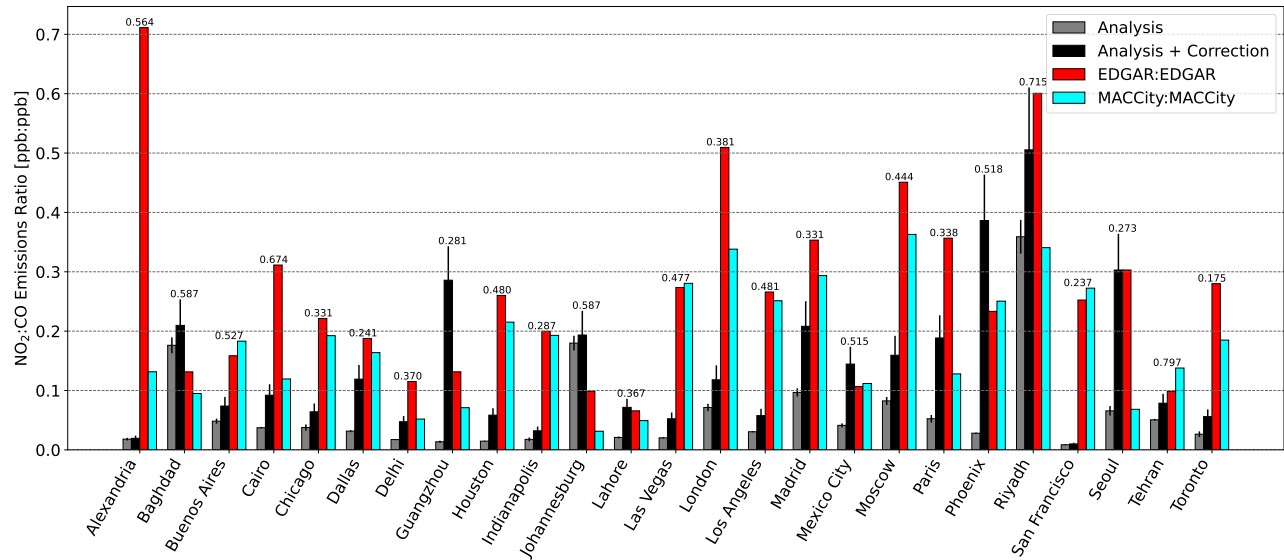

**Figure 8.** Same as in Figure 6 but for $NO_2$:CO ratios derived using observations from TROPOMI. EDGAR and MACCity are used to calculate inventory-based estimates of the emissions ratios.

these ratios. In the absence of cloud cover, we obtain 1–2 overpasses per day for each city. We consider a smaller subset of the TROPOMI product from June–August 2018, which is the same subset of data used by Lama et al. (2020), so that our enhancement ratios are derived from 23–86 overpasses for each city. Because there are more overpasses averaged into the $NO_2$:CO estimates of enhancement ratios, we expect them to be more robust than those involving OCO-2/3.

Figure 8 shows the derived $NO_2$:CO ratios compared to inventory-based estimates from EDGAR and MACCity. Here we observe ratios that are significantly lower than inventory-based estimates both before and after the lifetime correction has been applied. Again these differences are generally more significant over cities in North America and Europe, and Johannesburg and Baghdad are outliers, with observed enhancement ratios higher than either of the inventory estimates. Tables 2 & 3 summarize the results of all three sets of emissions ratios for all the cities considered. Figure 10 shows the median relative difference
between our enhancement ratios and those derived from inventories.

We also compare our $NO_2$:CO results to those of Lama et al. (2020) in Figure 9. Lama et al. applied two different methods to calculate $NO_2$:CO enhancement ratios in six megacities (Tehran, Mexico City, Cairo, Riyadh, Lahore and Los Angeles) using measurements from TROPOMI. Lama et al. also use an averaging kernel correction to their ratios, except with a different methodology; they apply column averaging kernels to reported profiles of $NO_2$ and CO from the Copernicus Atmospheric
Monitoring Service (CAMS) to determine the impact on $X_{NO_2}$ and $X_{CO}$. Lama et al. also apply a correction which accounts for the short lifetime of $NO_2$ due to chemical reactions with hydroxyl (OH) in the atmosphere by constructing a correcting scale factor using CAMS-reported OH concentrations and the observed wind speed. Comparing the results of Lama et al. (2020) to our calculated values without their respective $NO_2$ lifetime corrections, we find good agreement, within the uncertainties, in

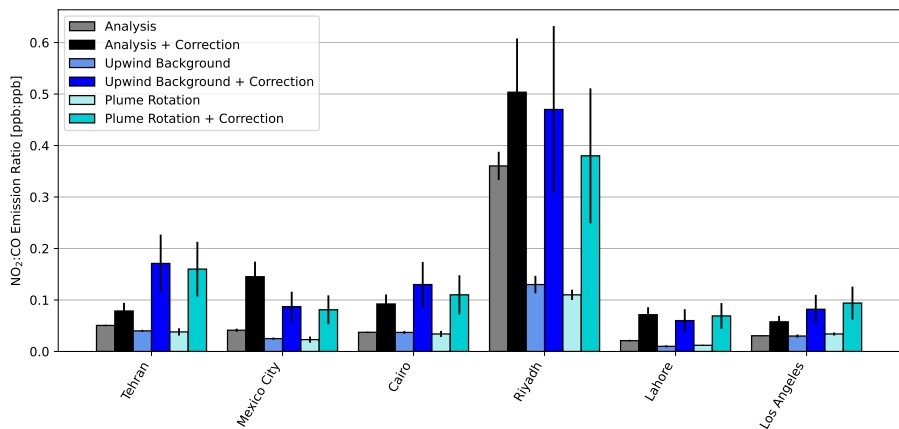

**Figure 9.** Comparison of our observed $NO_2$:CO ratios ("Analysis" and "Analysis + Correction") to those calculated by Lama et al. (2020). Lama et al. use two different methods to compute enhancements: their "Upwind Background" method and their "Plume Rotation" method. Both are shown here, with and without their respective lifetime corrections.

the ratios for all cities except Riyadh. After the application of the respective $NO_2$ lifetime corrections, all cities except Tehran show agreement within the uncertainties. Our $NO_2$ lifetime correction is significantly smaller in Tehran than the Lama et al. (2020) correction.

## 5 Discussion

### 5.1 Analysis of Measured Ratios

The $CO:CO_2$ ratios that we derive using TROPOMI and OCO-2/3 are larger than the inventory-based estimates calculated from EDGAR, ODIAC, and FFDAS in 71% of the cities we studied (Figure 6). This is the case for nearly all cities in North America and Europe, though mixed results are observed for the rest of the world; in cities such as Tehran (Iran) and Cairo (Egypt) we observe high ratios relative to inventory estimates, while in Johannesburg (South Africa), Baghdad (Iraq) and Buenos Aires (Argentina), we observe ratios that are lower than the inventory values. We also observe larger $CO:CO_2$ enhancement ratios relative to reported emissions ratios when using CO emissions from the MACCity inventory (Figure A1). Here, the low bias for inventory estimates appears to be even stronger for cities in North America and Europe, while Johannesburg and Baghdad remain as outliers with lower observed enhancement ratios.

For the $NO_2:CO_2$ ratios that we calculate (Figure 7), we find that without any correction for $NO_2$ lifetime, this trend is reversed: we observe ratios that are considerably lower than those derived from EDGAR, ODIAC and FFDAS in 91% of the cities. Upon application of our correction for $NO_2$ lifetime, the ratios for many of these cities are brought in closer agreement with the inventory estimates. Johannesburg is a notable outlier, with an observed ratio that is comparable to the EDGAR and FFDAS estimates both before and after correction, yet is around twice the EDGAR:ODIAC estimated ratio. Delhi, Mexico City

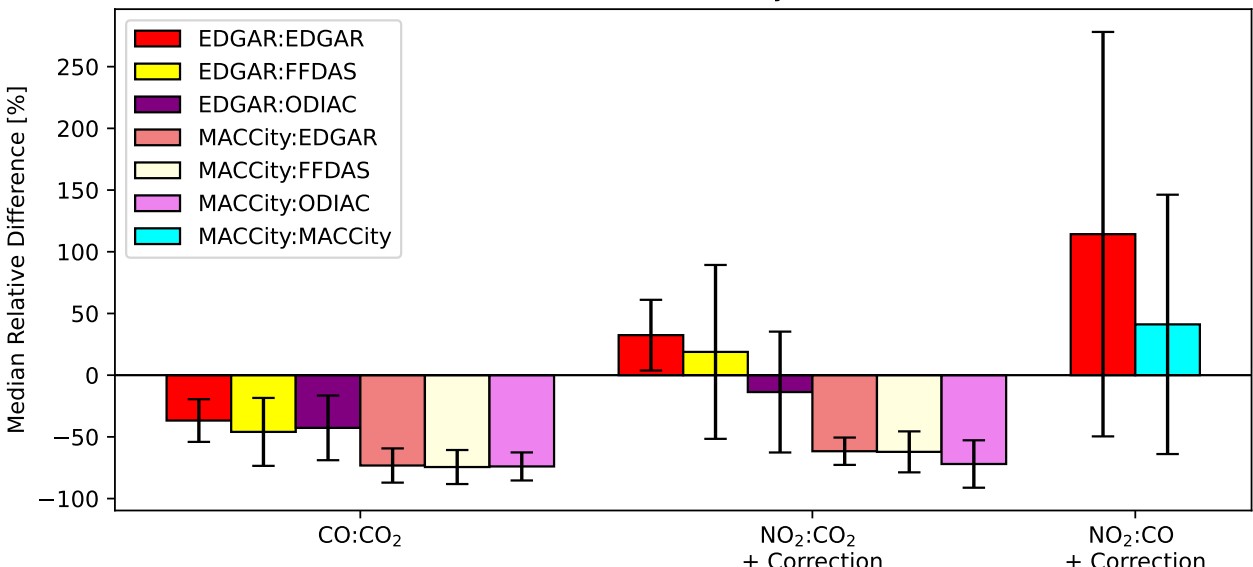

**Figure 10.** The median relative difference in percent between the observed enhancement ratios and those from the inventories (inventory-observed). The left-most set of bars is for the $CO:CO_2$ enhancement ratios. The middle set of bars is for lifetime-corrected $NO_2:CO_2$ enhancement ratios, and the right set of bars is for lifetime-corrected $NO_2:CO$ enhancement corrected. The error bars show the standard deviation in the spread in inventory differences from the measurements over all cities, divided by the square root of the number of cities.

and Tehran also have ratios that exceed inventory-based estimates by a similar amount after the lifetime correction has been applied. An added complication for the specific case of Johannesburg is the presence of a collection of large coal-fired power plants located ~100 km east of the city which together emit > 10 $TgCO_2$/yr and an additional two power plants located ~50 km to the south according to the Carbon Monitoring for Action (CARMA) database (Ummel, 2012). Plumes from these sites are clearly visible in CO and $NO_2$ measurements from TROPOMI and depending on the wind direction and distance from the city could influence the measurements in the OCO-2 swath. Interestingly, the positions of these four cities (Delhi, Johannesburg, Mexico City, Tehran) as outliers appears to be consistent for both EDGAR $NO_x$ emissions and those from MACCity.

In general, we find that measured $NO_2:CO$ enhancement ratios are smaller than those inferred from the inventories, while measured $CO:CO_2$ ratios are generally larger than those from the EDGAR, FFDAS, and ODIAC inventories (Figure 10). Measured $NO_2:CO_2$ ratios are in reasonable agreement with the inferred inventory enhancement ratios using EDGAR $NO_2$ (Figure 10). From our observed $CO:CO_2$ ratios, which are generally larger than the inventory ratios, we infer that the inventories we considered tend to either underestimate CO emissions, overestimate emissions of $CO_2$, or both. Similarly, the lifetime-corrected $NO_2:CO_2$ ratios that are observed to generally agree with the EDGAR $NO_x$-based estimates suggests that either the inventories accurately capture emissions of both $NO_2$ and $CO_2$, or that emissions of these two gases are both biased either high or low by a similar magnitude. Given that the spread provided by the three $CO_2$ inventories (which is on average

about 20% around the mean of the three, and exceeds 30% for only Alexandria, Delhi, Johannesburg and Moscow) coincides with the uncertainty range for the $NO_2$:$CO_2$ enhancement ratio in 8 out of 22 cities, and is close for another 7 cities, the $NO_2$:$CO_2$ emissions ratios do not appear to be affected by systematic biases as much as the $CO$:$CO_2$ ratios. From this, we infer that the discrepancies in the $NO_2$:$CO$ ratios are likely caused by an underestimation of $CO$ emissions within the EDGAR inventory. The MACCity $NO_2$:$CO$ ratios are also larger on average than the measured enhancements, but both $CO$:$CO_2$ and $NO_2$:$CO_2$ enhancement ratios are smaller than the measured enhancements (Figures 10, A1, and A2). This is consistent with the mean MACCity $CO$ emissions being lower than EDGAR $CO$ emissions by about 20% in the cities we have studied, while its mean $NO_2$ emissions are less than half of those reported by EDGAR in these cities (Table 1).

If we were to rescale the $CO$ inventory emissions so that their emissions ratios matched the observed $CO$:$CO_2$ enhancement ratios, EDGAR $CO$ emissions would have to be about doubled on average. For MACCity, the required rescaling factor is considerably higher; this was driven in part by the $CO$ emissions estimates in Riyadh (Saudi Arabia), Las Vegas (USA) and Phoenix (USA). This suggests that MACCity may systematically underestimate $CO$ emissions in desert cities. Even when neglecting these three cities, the low-biased emissions in MACCity would still require a rescaling factor of around 4 on average to match the observed enhancement ratios.

One potential source of error in this analysis is from biogenic emissions of $CO$ and cycling of $CO_2$, but we expect these effects will be small. Under the assumption that there is less vegetation within urban boundaries than outside the urban region, biogenic $CO$ and $CO_2$ emissions have the potential to affect the urban-rural gradients, especially during the growing season. According to recent studies, however, these gradients are significantly smaller than the enhancements we measure, suggesting that urban $CO$ and $CO_2$ enhancements are dominated by fossil fuel emissions (Plant et al., 2022b; Wu et al., 2022). Further modeling will be necessary to apply this kind of analysis to smaller emission sources.

A second possible source of error is the temporal representativeness of the satellite data used in this analysis. The overpasses that successfully pass our filtering criteria are biased toward sunnier conditions and are most often collected in summertime, and some sites have very few overpasses (e.g., Toronto). If the enhancement ratios change seasonally, as might be expected, this type of analysis could cause a representativeness error, in which the comparisons between the measured enhancement ratios and the reported annual inventory ratios are systematically biased. Currently, the EDGAR and MACCity inventories, which provide $CO$ and $NO_2$ emissions, do not report sub-annual emissions, so comparing to seasonal inventory ratios is not possible. With longer satellite time series providing more opportunities for wintertime enhancement ratios, we will be able to compute robust annual enhancement ratios to compare with the annual inventories.

## 5.2 Emissions Estimates using High-Resolution Inventories

In addition to the global, gridded inventories that we have employed up to this point, the cities of Indianapolis (USA) and Los Angeles (USA) also have high-resolution anthropogenic $CO_2$ inventories. The Hestia Inventory provides gridded $CO_2$ fluxes for the cities of Los Angeles, Indianapolis, Salt Lake City and Baltimore at both hourly and annual temporal resolutions for the years 2010–2015 (Gurney et al., 2018b, 2019). The inventory for Los Angeles is provided at a spatial resolution of 1 km $\times$ 1 km for the SoCAB and the surrounding area, and the inventory for Indianapolis is given on a 200 m $\times$ 200 m grid. When summed

across the GHS polygons for each city, the annual emissions for Los Angeles and Indianapolis from the Hestia inventory are 120.9 $TgCO_2$/yr and 13.5 $TgCO_2$/yr, respectively. For Los Angeles, the annual estimates of the global gridded inventories (EDGAR, ODIAC, and FFDAS) are between 25-33% lower than this high resolution estimate, while for Indianapolis, the Hestia estimate is similar to the mean of the three global inventories. When comparing with the TIMES-corrected emissions

rates from the global inventories for Los Angeles, the estimates are brought into better agreement with the Hestia inventory, with emissions rates that are now only 6-17% lower than Hestia. For Indianapolis, the TIMES-corrected EDGAR estimate of 9.5 $TgCO_2$/yr is about 30% lower than the Hestia estimate, while the ODIAC and FFDAS values are around 30% higher.

     Using the $CO:CO_2$ enhancement ratio that was calculated for Los Angeles along with the Hestia $CO_2$ emissions estimate and equation 1, we estimate CO emissions to be $635 \pm 127$ GgCO/yr after assuming a 20% uncertainty in the Hestia emissions

estimate, which has good overlap with the estimate of $487 \pm 122$ GgCO/yr found by Hedelius et al. (2018) for 2013–2016, as well as the value of 581 GgCO/yr for the SoCAB which is reported by the California Air Resources Board (CARB) for the year 2015 in the CARB2017 database (https://www.arb.ca.gov/app/emsinv/2017/emssumcat.php). CARB2017 projections for the year 2020 estimate that SoCAB emissions of CO should decrease by 21%. The EDGAR and MACCity CO emissions estimates for Los Angeles are significantly lower at 301.4 and 104.7 GgCO/yr, respectively (Table 1).

In a similar way, we estimate emissions of $NO_2$ within the SoCAB to be $89 \pm 17$ $GgNO_2$/yr (after the $NO_2$ lifetime correction is applied), which agrees with the CARB estimate for 2015 of 105 $GgNO_2$/yr. CARB2017 projections for the year 2020 estimate that SoCAB emissions of $NO_x$ should decrease by 26% respectively. However, our estimated emissions are smaller than the annual EDGAR estimate of 132 $GgNO_2$/yr and larger than the MACCity estimate of 43.3 $GgNO_2$/yr.

     A similar approach using the Hestia $CO_2$ Inventory for Indianapolis yields estimated CO emissions of $115 \pm 23$ GgCO/yr,

much higher than both the EDGAR and MACCity estimates of 40.7 GgCO/yr and 11.7 GgCO, respectively. Using the lifetime-corrected $NO_2:CO_2$ ratio, emissions of $NO_2$ are estimated to be $7.8 \pm 1.6$ $GgNO_2$/yr, which is considerably higher than the MACCity estimate of 3.9 $GgNO_2$/yr, but lower than the the EDGAR estimate of 13.4 $GgNO_2$/yr.

     These investigations of Los Angeles and Indianapolis illustrate how the high MACCity $NO_2:CO$ ratios observed in Figure 8 are driven by a strong underestimation of CO emissions, even though the $NO_x$ emissions are also underestimated in MACCity

compared to the estimate derived from enhancement ratios and Hestia.

## 6   Conclusions

This study demonstrates a method to derive enhancement ratios between $CO_2$, CO and $NO_2$ using measurements from the OCO-2, OCO-3, and TROPOMI satellite instruments located downwind of or over large urban areas. This method is applied to derive enhancement ratios for 27 cities from around the world. These ratios are then compared to enhancement ratios derived

from the EDGAR, ODIAC, FFDAS, and MACCity global inventories. We find that $CO:CO_2$ ratios from these inventories are generally lower in cities across Europe and North America compared to the satellite-based ratios. After applying a correction to account for the short atmospheric lifetime of $NO_2$, observed $NO_2:CO_2$ ratios are mostly higher than inventory ratios when using $NO_x$ emissions from MACCity but generally show good agreement when using emissions from EDGAR, apart from

a few outlier cities where observed ratios were high compared to inventory estimates. Lifetime-corrected $NO_2$:CO ratios retrieved from TROPOMI observations over these cities show low values relative to inventory estimates and good agreement with the lifetime-corrected $NO_2$:CO ratios derived in a previous study by Lama et al. (2020).

We demonstrate that deriving enhancement ratios between more than two species can aid in the interpretation of results. By measuring ratios of CO:$CO_2$, $NO_2$:$CO_2$, and $NO_2$:CO, we are able to better diagnose which emissions lead to discrepancies between satellite- and inventory-derived ratios. For the EDGAR inventory, this analysis suggests an underestimation of CO emissions by around 50% on average, while for the MACCity inventory, we infer a more significant underestimation of CO emissions of about 75% on average, alongside a smaller underestimation of $NO_x$ emissions. In both EDGAR and MACCity, many of the largest underestimations of CO are observed for cities in Europe and North America, with MACCity showing significant underestimation in desert cities (Riyadh, Phoenix, Las Vegas). Further, we show that by combining these enhancement ratios with high-resolution $CO_2$ inventories, emissions of CO and $NO_2$ can be calculated, which, in the case of Los Angeles, show good agreement with both region-specific inventories and previous modelling studies. These analyses with high-resolution inventories additionally provide further support for the underestimation of urban CO emissions in EDGAR and MACCity.

There is considerable potential for further study using the methodology that has been laid out here. In particular, these methods could be applied to other anthropogenic co-emitters of $CO_2$, CO and $NO_2$. Fossil fuel burning power plants are a candidate for future investigations, as other studies have already used multi-sensor techniques involving $NO_2$ and $CO_2$ to estimate power plant emissions (e.g., Reuter et al., 2019; Hakkarainen et al., 2021). Furthermore, enhancement ratios involving other species are observed by TROPOMI, such as $CH_4$, HCHO, and $SO_2$ over urban regions could be explored using the framework that has been described here.

Due to the limited number of usable co-locations between OCO-2/3 and TROPOMI that were available in this study, we have limited our enhancement ratio results to single values across the full time periods. As the constellation of $CO_2$ observing satellites expands in the coming years, there will be greater potential for co-locations of observations, which could provide reliable information on long-term trends of these enhancement ratios and open up the possibility for comparison to trends in ratios derived from emissions inventories. When paired with state-of-the-art $CO_2$ inventories, these enhancement ratios could provide a flexible framework to determine whether emissions reduction targets for a wide array of greenhouse gases and pollutants are being met on schedule by cities around the world.

*Data availability.* OCO-2 and OCO-3 data were obtained from the Goddard DAAC https://disc.gsfc.nasa.gov/datasets/OCO2_L2_Lite_FP_9r/summary, https://disc.gsfc.nasa.gov/datasets/OCO3_L2_Lite_FP_10.4r/summary?keywords=OCO-3%20vEarly. TROPOMI data were obtained from the NASA GES DISC (CO: https://tropomi.gesdisc.eosdis.nasa.gov/data/S5P_TROPOMI_Level2/S5P_L2__CO_____.1 and $NO_2$: https://tropomi.gesdisc.eosdis.nasa.gov/data/S5P_TROPOMI_Level2/S5P_L2__NO2____HiR.1/. TROPOMI $NO_2$ data for the lifetime calculation was obtained from the Sentinel-5P hub (https://s5phub.copernicus.eu/dhus/). A priori $NO_2$ profiles from the TM5 model were obtained from the Sentinel-5P hub (https://s5phub.copernicus.eu/dhus/). GHS-UCDB is available from https://data.europa.eu/doi/10.2760/

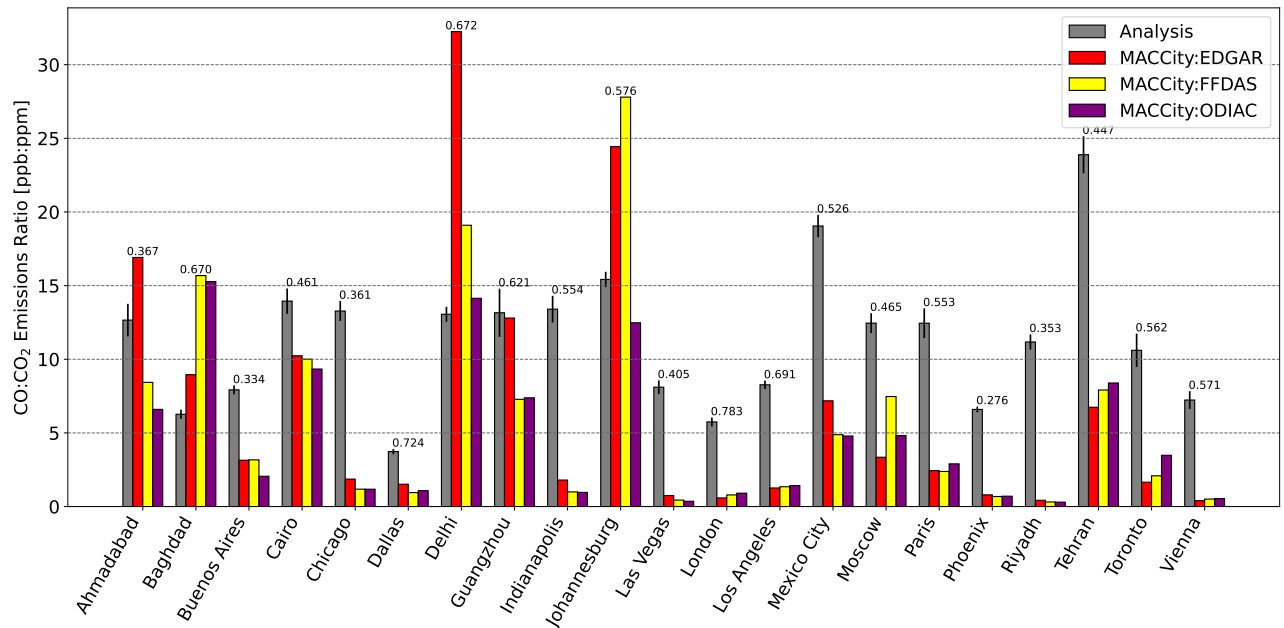

**Figure A1.** Same as is Figure 6 but for enhancement ratios calculated using CO emissions from the MACCity inventory.

037310. ODIAC2018 is available from http://doi.org/10.17595/20170411.001. FFDAS2.2 is available from http://ffdas.rc.nau.edu/Data.html. TIMES is available from https://cdiac.ess-dive.lbl.gov/ftp/Nassar_Emissions_Scale_Factors/. EDGAR5.0 is available from https://edgar.jrc. ec.europa.eu/dataset_ghg50. MACCity is available from http://accent.aero.jussieu.fr/MACC_metadata.php. MERRA-2 data are available from https://gmao.gsfc.nasa.gov/reanalysis/MERRA-2/. The Hestia $CO_2$ inventory databases are available from https://catalog.data.gov/ dataset/hestia-fossil-fuel-carbon-dioxide-emissions-inventory-for-urban-regions-82c05.

## Appendix A:  Comparison of $CO:CO_2$ and $NO_2:CO_2$ ratios with emission ratios derived from MACCity

Comparison of our observed $CO:CO_2$ enhancement ratios with emission ratios calculated using CO emissions from the MAC-City inventory are shown in Figure A1. Results are similar to those using the EDGAR CO emissions, with underestimation by the inventories relative to the observations even more pronounced for many cities in Europe and North America. Similarly, Figure A2 shows a comparison of observed $NO_2:CO_2$ with MACCity derived emissions ratios. These emission ratios are characterized by a greater underestimation relative to the observed enhancement ratios when compared with the EDGAR emissions. As with the EDGAR emissions ratios, the cities of Delhi, Johannesburg, Mexico City and Tehran stand out as the cases where this underestimation is the most pronounced.

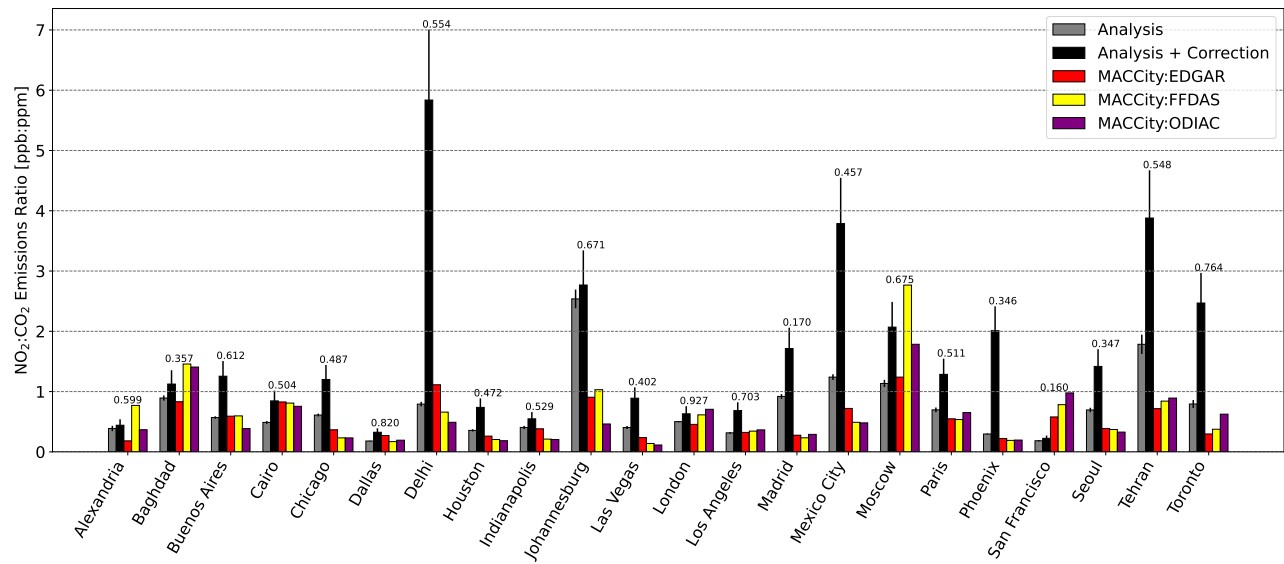

**Figure A2.** Same as in Figure 7 but for enhancement ratios calculated using $NO_x$ emissions from the MACCity inventory.

## Appendix B: $NO_2$ lifetime calculation

We compute $NO_2$ lifetimes similarly to Laughner and Cohen (2019) using $NO_2$ column densities from offline TROPOMI data (processor version 1.3). Wind direction for each day is calculated from GEOS-5 FP-IT reanalysis data (Lucchesi, 2015) by interpolating the bottom five levels of the wind fields to 13:30 local time. Horizontal averaging uses a flat topped Gaussian (fourth power) centered on each city, with a width chosen based on the city size. $NO_2$ column densities from each day are rotated so that the wind directions are aligned, and pixels with `qa_value > 0.75` are averaged in time, weighted by the pixel area. Line densities are computed by integrating the rotated line densities perpendicular to the wind direction. An exponentially-modified Gaussian function,

$$F(x|a, x_0, \mu_x, \sigma_x, B) = \frac{a}{2x_0} \exp\left(\frac{\mu_x}{x_0} + \frac{\sigma_x^2}{2x_0^2} - \frac{x}{x_0}\right) \text{erfc}\left(-\frac{1}{\sqrt{2}}\left[\frac{x - \mu_x}{\sigma_x} - \frac{\sigma_x}{x_0}\right]\right) \tag{B1}$$

is fit to the line densities. $x$ is the along wind distance and $a$, $x_0$, $\mu_x$, $\sigma_x$, and $B$ are fitting parameters. $\text{erfc}$ is the error function complement. Lifetime is calculated as $x_0/\overline{u}$, where $\overline{u}$ is the average wind speed from GEOS FP-IT.

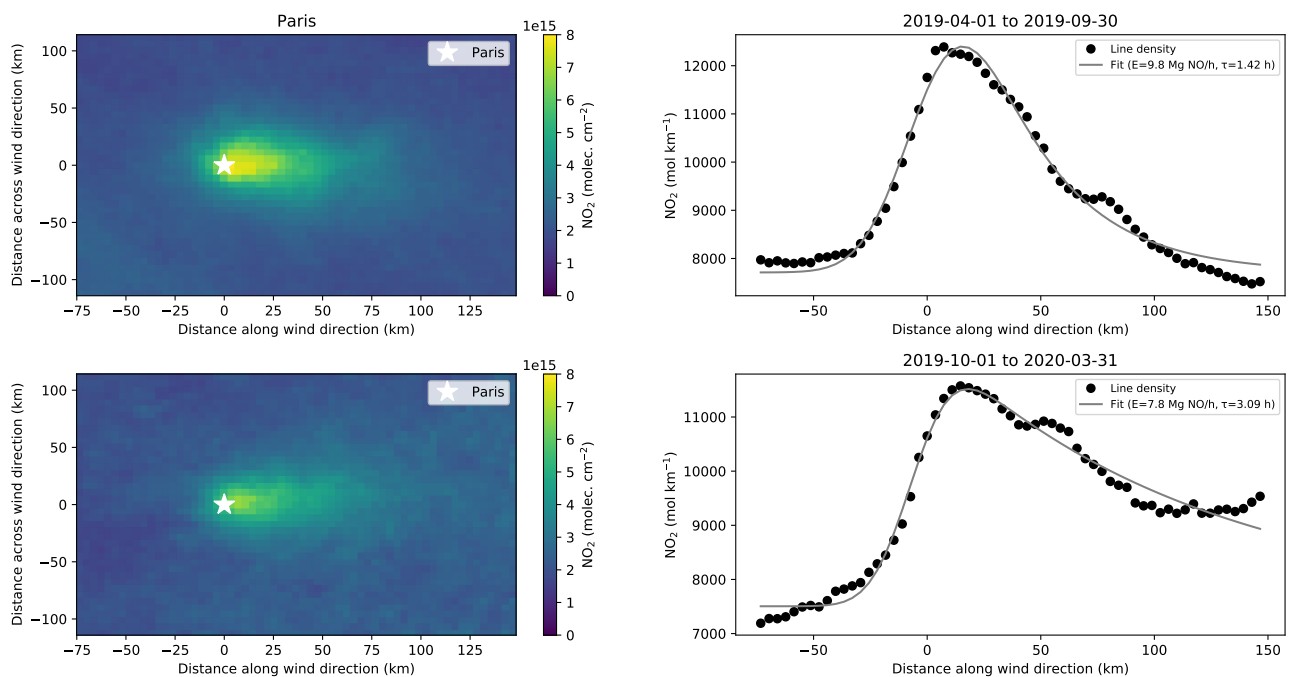

**Figure B1.** Example of lifetime fitting using Paris as a demonstration. The top row shows summer fits and the bottom row winter fits, with the specific dates used given in the title of the right panels. The left columns show the average NO₂ column density after aligning wind direction (right = downwind). The right columns show the line densities (black circles) and fits (grey lines) computed from the wind-aligned column densities.

**Table 1.** Summary of inventory-based emissions estimates for all the cities that were considered. Gridded emissions are summed across the extent of the GHS polygon for each city. For the ODIAC and MACCity inventories, which are provided at a monthly temporal resolution, the estimate is the mean across all 12 months.

| City | 2015 Population [10^6] | ODIAC $CO_2$ [Tg/yr] | FFDAS $CO_2$ [Tg/yr] | EDGAR $CO_2$ [Tg/yr] | EDGAR CO [Gg/yr] | MACCity CO [Gg/yr] | EDGAR $NO_2$ [Gg/yr] | MACCity $NO_2$ [Gg/yr] |
|---|---|---|---|---|---|---|---|---|
| Ahmadabad | 6.7 | 15.3 | 12.5 | 6.2 | 68.0 | 58.6 | 15.3 | 4.3 |
| Alexandria | 5.5 | 8.9 | 4.9 | 19.6 | 29.2 | 27.7 | 36.2 | 3.9 |
| Baghdad | 5.4 | 10.7 | 10.0 | 17.4 | 292.5 | 130.0 | 65.4 | 18.9 |
| Buenos Aires | 13.9 | 45.8 | 28.8 | 29.9 | 316.5 | 76.7 | 80.8 | 23.7 |
| Cairo | 19.7 | 38.9 | 39.7 | 37.9 | 122.3 | 274.1 | 66.0 | 35.4 |
| Chicago | 6.8 | 80.8 | 80.0 | 51.1 | 217.2 | 73.8 | 78.4 | 23.9 |
| Dallas | 5.2 | 42.9 | 49.2 | 30.8 | 171.2 | 41.7 | 52.7 | 11.9 |
| Delhi | 26.7 | 97.0 | 68.1 | 41.2 | 549.4 | 889.1 | 100.9 | 53.8 |
| Guangzhou | 40.6 | 302.9 | 300.6 | 177.1 | 1371.0 | 1542.1 | 295.5 | 166.8 |
| Houston | 4.9 | 57.0 | 54.6 | 43.3 | 161.3 | 37.7 | 70.4 | 14.3 |
| Indianapolis | 1.1 | 15.2 | 14.5 | 7.8 | 40.7 | 11.3 | 13.4 | 3.9 |
| Johannesburg | 6.5 | 43.7 | 19.6 | 22.5 | 404.6 | 339.1 | 66.4 | 22.4 |
| Lahore | 10.1 | 9.7 | 6.0 | 6.3 | 252.9 | 128.3 | 28.0 | 6.7 |
| Las Vegas | 2.0 | 20.6 | 16.7 | 10.0 | 42.3 | 6.6 | 19.0 | 3.3 |
| London | 9.6 | 17.4 | 23.1 | 31.2 | 49.7 | 16.9 | 42.0 | 14.7 |
| Los Angeles | 14.3 | 80.8 | 84.1 | 89.9 | 301.4 | 104.7 | 132.0 | 43.3 |
| Madrid | 4.9 | 9.1 | 11.0 | 9.5 | 27.9 | 5.5 | 16.2 | 4.2 |
| Mexico City | 19.6 | 40.0 | 38.4 | 26.1 | 362.8 | 129.9 | 63.5 | 21.8 |
| Moscow | 14.1 | 79.9 | 46.9 | 110.4 | 212.7 | 263.7 | 160.1 | 170.8 |
| Paris | 9.7 | 22.4 | 26.4 | 25.2 | 50.5 | 58.3 | 29.5 | 19.5 |
| Phoenix | 3.6 | 27.1 | 26.9 | 23.1 | 100.9 | 15.0 | 38.3 | 6.7 |
| Riyadh | 5.7 | 101.7 | 92.8 | 68.5 | 136.4 | 22.8 | 131.0 | 13.0 |
| San Francisco | 4.6 | 13.8 | 16.4 | 22.7 | 85.9 | 38.6 | 35.6 | 17.3 |
| Seoul | 21.6 | 122.4 | 99.8 | 96.6 | 286.6 | 380.5 | 141.7 | 47.0 |
| Tehran | 12.5 | 46.8 | 49.1 | 57.7 | 654.2 | 294.4 | 106.1 | 50.8 |
| Toronto | 6.0 | 27.4 | 41.2 | 53.1 | 136.8 | 68.7 | 63.0 | 20.0 |
| Vienna | 1.9 | 6.1 | 6.2 | 8.1 | 12.3 | 2.6 | 10.2 | 1.5 |

**Table 2.** All enhancement ratios derived using OCO-2/3 and TROPOMI. Overpasses must individually have sufficient linear dependence (R> 0.2). Cities without any such overpasses are marked by a dash.

| City | No. of OCO-2 / 3 Overpasses | $CO:CO_2$ Enhancement Ratio [ppb:ppm] | No. of OCO-2 / 3 Overpasses | $NO_2:CO_2$ Enhancement Ratio [ppb:ppm] | No. of TROPOMI Overpasses | $NO_2:CO$ Enhancement Ratio [ppb:ppb] |
|---|---|---|---|---|---|---|
| Ahmadabad | 3 | $12.7 \pm 1.1$ | - | - | - | - |
| Alexandria | - | - | 5 | $0.39 \pm 0.04$ | 30 | $0.018 \pm 0.002$ |
| Baghdad | 2 / 2 | $6.3 \pm 0.3$ | 6 / 2 | $0.89 \pm 0.04$ | 67 | $0.176 \pm 0.013$ |
| Buenos Aires | 6 / 4 | $7.9 \pm 0.3$ | 6 / 4 | $0.57 \pm 0.02$ | 39 | $0.048 \pm 0.004$ |
| Cairo | 12 / 2 | $14.0 \pm 0.9$ | 12 / 2 | $0.49 \pm 0.02$ | 84 | $0.037 \pm 0.001$ |
| Chicago | 5 | $13.3 \pm 0.7$ | 5 | $0.61 \pm 0.03$ | 44 | $0.038 \pm 0.005$ |
| Dallas | 3 / 1 | $3.7 \pm 0.2$ | 3 / 1 | $0.18 \pm 0.01$ | 57 | $0.032 \pm 0.002$ |
| Delhi | 3 / 1 | $13.0 \pm 0.5$ | 4 / 1 | $0.79 \pm 0.04$ | 46 | $0.017 \pm 0.001$ |
| Guangzhou | 3 | $13.2 \pm 1.6$ | - | - | 23 | $0.014 \pm 0.002$ |
| Houston | - | - | 5 / 1 | $0.36 \pm 0.02$ | 54 | $0.015 \pm 0.001$ |
| Indianapolis | 7 | $13.2 \pm 0.9$ | 7 | $0.41 \pm 0.02$ | 32 | $0.018 \pm 0.003$ |
| Johannesburg | 11 | $15.4 \pm 0.5$ | 11 | $2.54 \pm 0.16$ | 62 | $0.180 \pm 0.012$ |
| Lahore | - | - | - | - | 42 | $0.021 \pm 0.002$ |
| Las Vegas | 10 / 1 | $8.1 \pm 0.5$ | 10 / 1 | $0.41 \pm 0.02$ | 58 | $0.021 \pm 0.001$ |
| London | 1 / 1 | $5.7 \pm 0.3$ | 1 / 1 | $0.50 \pm 0.01$ | 47 | $0.071 \pm 0.006$ |
| Los Angeles | 7 / 2 | $8.2 \pm 0.3$ | 7 / 2 | $0.32 \pm 0.01$ | 84 | $0.031 \pm 0.001$ |
| Madrid | - | - | 9 / 1 | $0.92 \pm 0.04$ | 49 | $0.097 \pm 0.007$ |
| Mexico City | 7 / 5 | $19.1 \pm 0.7$ | 7 / 5 | $1.24 \pm 0.05$ | 50 | $0.041 \pm 0.003$ |
| Moscow | 5 | $12.5 \pm 0.7$ | 5 | $1.13 \pm 0.06$ | 53 | $0.082 \pm 0.007$ |
| Paris | 4 / 2 | $12.4 \pm 1.0$ | 4 / 2 | $0.70 \pm 0.04$ | 39 | $0.052 \pm 0.007$ |
| Phoenix | 14 / 6 | $6.6 \pm 0.2$ | 14 / 6 | $0.30 \pm 0.01$ | 74 | $0.028 \pm 0.001$ |
| Riyadh | 8 | $11.2 \pm 0.5$ | - | - | 52 | $0.359 \pm 0.030$ |
| San Francisco | - | - | 7 | $0.18 \pm 0.01$ | 62 | $0.009 \pm 0.001$ |
| Seoul | - | - | 4 | $0.698 \pm 0.04$ | 45 | $0.066 \pm 0.008$ |
| Tehran | 10 / 3 | $23.9 \pm 1.2$ | 10 / 3 | $1.78 \pm 0.16$ | 86 | $0.051 \pm 0.001$ |
| Toronto | 1 | $10.6 \pm 1.2$ | 1 | $0.80 \pm 0.07$ | 39 | $0.026 \pm 0.005$ |
| Vienna | 2 | $7.2 \pm 0.6$ | - | - | - | - |

**Table 3.** All $NO_2$:$CO_2$ and $NO_2$:CO enhancement ratios derived using OCO-2/3 and TROPOMI, including $NO_2$ lifetime correction. Overpasses must individually have sufficient linear dependence (R> 0.2). Cities without any such overpasses are marked by a dash.

| City | No. of OCO-2 / 3 Overpasses | Average Wind Speed for Correction (m/s) | $NO_2$:$CO_2$ Enhancement Ratio [ppb:ppm] | No. of TROPOMI Overpasses | Average Wind Speed for Correction (m/s) | $NO_2$:CO Enhancement Ratio [ppb:ppb] |
|---|---|---|---|---|---|---|
| Alexandria | 5 | 5.13 | 0.44 ± 0.10 | 30 | 6.30 | 0.019 ± 0.005 |
| Baghdad | 6 / 2 | 5.79 | 1.12 ± 0.23 | 67 | 6.91 | 0.210 ± 0.044 |
| Buenos Aires | 6 / 4 | 4.33 | 1.26 ± 0.25 | 39 | 6.58 | 0.074 ± 0.015 |
| Cairo | 12 / 2 | 4.39 | 0.85 ± 0.17 | 84 | 5.02 | 0.092 ± 0.018 |
| Chicago | 5 | 4.49 | 1.20 ± 0.24 | 44 | 4.75 | 0.064 ± 0.014 |
| Dallas | 3 / 1 | 4.41 | 0.33 ± 0.07 | 57 | 4.37 | 0.119 ± 0.024 |
| Delhi | 4 / 1 | 2.15 | 5.83 ± 1.17 | 46 | 4.65 | 0.047 ± 0.010 |
| Guangzhou | - | - | - | 23 | 3.88 | 0.286 ± 0.057 |
| Houston | 5 / 1 | 4.94 | 0.74 ± 0.15 | 56 | 3.84 | 0.058 ± 0.012 |
| Indianapolis | 7 | 4.45 | 0.55 ± 0.11 | 32 | 4.29 | 0.032 ± 0.007 |
| Johannesburg | 11 | 5.20 | 2.77 ± 0.58 | 62 | 5.55 | 0.193 ± 0.041 |
| Lahore | - | - | - | 42 | 2.73 | 0.072 ± 0.014 |
| Las Vegas | 10 / 1 | 4.44 | 0.89 ± 0.18 | 58 | 3.87 | 0.052 ± 0.011 |
| London | 1 / 1 | 6.86 | 0.63 ± 0.13 | 47 | 4.38 | 0.118 ± 0.024 |
| Los Angeles | 7 / 2 | 3.59 | 0.68 ± 0.14 | 84 | 4.34 | 0.058 ± 0.012 |
| Madrid | 9 / 1 | 5.22 | 1.71 ± 0.34 | 49 | 3.92 | 0.208 ± 0.042 |
| Mexico City | 7 / 5 | 2.80 | 3.79 ± 0.76 | 50 | 2.56 | 0.145 ± 0.029 |
| Moscow | 5 | 5.92 | 2.08 ± 0.42 | 53 | 5.42 | 0.159 ± 0.033 |
| Paris | 4 / 2 | 5.44 | 1.29 ± 0.26 | 39 | 3.51 | 0.188 ± 0.038 |
| Phoenix | 14 / 6 | 4.17 | 2.01 ± 0.41 | 74 | 3.59 | 0.386 ± 0.078 |
| Riyadh | - | - | - | 52 | 7.30 | 0.505 ± 0.105 |
| San Francisco | 7 | 4.73 | 0.22 ± 0.05 | 62 | 4.80 | 0.010 ± 0.002 |
| Seoul | 4 | 7.57 | 1.42 ± 0.29 | 45 | 3.55 | 0.303 ± 0.061 |
| Tehran | 10 / 3 | 3.26 | 3.89 ± 0.79 | 86 | 5.28 | 0.079 ± 0.016 |
| Toronto | 1 | 2.88 | 2.47 ± 0.50 | 39 | 4.29 | 0.056 ± 0.012 |

## Appendix C: Column averging kernel corrections

To compute accurate surface enhancements, we need to take into account the sensitivity of the measurement to changes in trace gas concentrations near the surface. In previous work, a simple scaling of the measured anomalies by the surface pressure averaging kernel is performed (e.g., Wunch et al., 2009), which is a valid approach when the a priori enhancement between the source region and background region is zero and the averaging kernels do not vary spatially, but that approach is not generally applicable. In this appendix, we will derive the general case, describe some simplifying assumptions, and identify the correct approach for retrievals with a priori profiles that vary spatially, like for the TROPOMI $NO_2$ and CO retrievals.

Starting with Rodgers and Connor (2003) equation 4, we can write down that the retrieved profile, $\hat{\mathbf{x}}$ is related to the true profile, $\mathbf{x^t}$ through a smoothing by the averaging kernel matrix, $\mathbf{A}$, and the a priori profile, $\mathbf{x^a}$. Integrating both sides of this equation using the pressure weighting function, $\mathbf{h^T}$, produces the same equation but for the total column, $c$ (similar to Rodgers and Connor equation 22).

$$\hat{\mathbf{x}} = \mathbf{x^a} + \mathbf{A}(\mathbf{x^t} - \mathbf{x^a}) \tag{C1}$$

$$\hat{c} = c^a + \mathbf{h^T}\mathbf{A}(\mathbf{x^t} - \mathbf{x^a}) \tag{C2}$$

The $\mathbf{h^T}$ vector is the pressure weighting function, such that $\mathbf{h^T}\mathbf{x} = c$, and the column averaging kernel, $\mathbf{a}$, is defined by Connor et al. (2008)'s equation 8 as $a_j = (\mathbf{h^T}\mathbf{A})_j \frac{1}{h_j}$, such that $\mathbf{h^T}\mathbf{A}(\mathbf{x^t} - \mathbf{x^a}) = \sum_j a_j h_j (x_j^t - x_j^a)$.

In our analyses, we collect two sets of total column measurements: one inside and one outside the urban plume. The measurements will be called $\hat{c}_u$ and $\hat{c}_b$ for urban and background, respectively. Each will have its column averaging kernel, $a_u$ and $a_b$, its prior profile, $x_u^a$ and $x_b^a$, and its prior column, $c_u^a$ and $c_b^a$.

$$\hat{c}_u = c_u^a + \sum_j a_{ju} h_j (x_{ju}^t - x_{ju}^a) \tag{C3}$$

$$\hat{c}_b = c_b^a + \sum_j a_{jb} h_j (x_{jb}^t - x_{jb}^a) \tag{C4}$$

We are interested in finding the true enhancement between the urban and background columns, i.e., $\Delta c^t = c_u^t - c_b^t$, which we assume is entirely constrained to the surface layer. Computing the measured anomalies using equations C3 and C4, we get:

$$\begin{aligned} \Delta\hat{c} &= \hat{c}_u - \hat{c}_b \\ &= c_u^a + \sum_j a_{ju} h_j (x_{ju}^t - x_{ju}^a) - c_b^a - \sum_j a_{jb} h_j (x_{jb}^t - x_{jb}^a) \\ &= \Delta c^a + \sum_j \left( a_{ju} h_j (x_{ju}^t - x_{ju}^a) - a_{jb} h_j (x_{jb}^t - x_{jb}^a) \right) \end{aligned} \tag{C5}$$

where $\Delta c^a = c_u^a - c_b^a$.

To isolate the true column difference in equation C5, we need to make an assumption, as it is otherwise intractable. We can assume any combination of the following under the correct conditions:

1. The averaging kernels are the same inside and outside the plume: $\mathbf{a_u} = \mathbf{a_b} = \mathbf{a}$

2. The priors are the same inside and outside the plume: $c_u^a = c_b^a = c^a$, and $\mathbf{x_u^a = x_b^a = x^a}$

3. The a prioris are perfect: $c_u^t = c_u^a$, $c_b^t = c_b^a$, $\mathbf{x_u^t = x_u^a}$, and $\mathbf{x_b^t = x_b^a}$

If we can reasonably assume that the averaging kernels are the same inside and outside the plume ($\mathbf{a_u = a_b = a}$), then we can simplify equation C5:

$$
\begin{aligned}
\Delta\hat{c} &= \Delta c^a + \sum_j (a_j h_j (x_{ju}^t - x_{jb}^t) - a_j h_j (x_{ju}^a - x_{jb}^a) \\
&= \Delta c^a + \sum_j a_j h_j (\Delta x_j^t - \Delta x_j^a) \tag{C6}
\end{aligned}
$$

If we can reasonably assume that the priors are the same inside and outside the plume, then equation C5 simplifies, but remains intractable:

$$
\Delta\hat{c} = \sum_j \left( a_{ju} h_j x_{ju}^t - a_{ju} h_j x_j^a - a_{jb} h_j x_j^t + a_{jb} h_j x_j^a \right) \tag{C7}
$$

However, if we assume that $\mathbf{a_u = a_b = a}$ and $\mathbf{x_u^a = x_b^a = x^a}$, this simplifies further:

$$
\begin{aligned}
\Delta\hat{c} &= \sum_j \left( a_j h_j x_{ju}^t - a_j h_j x_j^a - a_j h_j x_{jb}^t + a_j h_j x_j^a \right) \\
&= \sum_j a_j h_j (x_{ju}^t - x_{jb}^t) \tag{C8}
\end{aligned}
$$

The equation above is what is often assumed, and rearranging equation C8 to solve for the true enhancement ($\Delta c^t = c_u^t - c_b^t$) gives equation C9, under the assumption that the enhancement is constrained to the surface layer, and therefore $\Delta x_j^t = 0, \forall j > 1$. We therefore need only the value of the column averaging kernel at the surface, which we will call $a^0$, and we recall that $\sum_j h_j x_j = c$ to write:

$$
\Delta c^t = \frac{\Delta\hat{c}}{a^0} \tag{C9}
$$

For the OCO-2 and OCO-3 retrievals, the averaging kernels and a priori profiles inside and outside the plume are negligibly different, and so equation C9 is the correct one to use. For the TROPOMI $NO_2$ and CO retrievals, we cannot assume that the a priori profiles are the same inside and outside the plume. However, the reported averaging kernels do not differ significantly inside and outside the urban plume, so it is appropriate to use equation C6 under most conditions. (Exceptions to this might be found, for example, in cities with significantly different surface albedo compared with the surrounding region.) Starting with equation C6, again assuming that the enhancements are constrained to the surface layer, we can isolate $\Delta c^t$ and rearrange:

$$
\begin{aligned}
\Delta\hat{c} &= \Delta c^a + \sum_j a_j h_j (\Delta x_j^t - \Delta x_j^a) \\
&= \Delta c^a + a^0 (\Delta c^t - \Delta c^a) \\
\Delta c^t &= \frac{\Delta\hat{c}}{a^0} - \frac{(1 - a^0)\Delta c^a}{a^0} \tag{C10}
\end{aligned}
$$

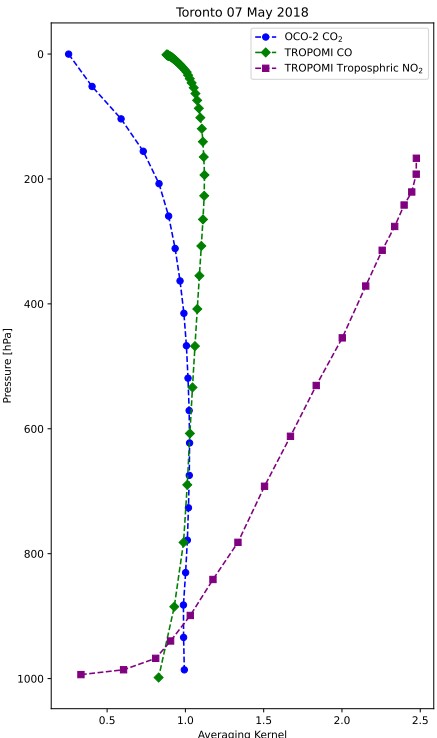

**Figure C1.** Averaging kernel profiles for OCO-2 $CO_2$, TROPOMI CO, and TROPOMI $NO_2$ at Toronto enhancement on 7 May 2018.

This equation includes an extra correction term compared with equation C9 that adjusts the measured anomalies by the difference between the urban and rural priors (the prior enhancement), weighted by their contribution to the retrieval $(1 - a^0)$, and divided by the surface pressure averaging kernel. If $c_u^a = c_b^a$, then equation C10 reduces to equation C9.

We include sample averaging kernel profiles for reference (Figure C1) and an example of distribution of surface averaging kernel values in the enhancement/background (Figure C2).

*Author contributions.* CGM designed the study, performed data analysis and wrote the manuscript; JPM implemented the referee suggestions and updated the manuscript. DW provided guidance across all stages of the process. JKH and RN supplied code which became part of the data analysis for the study and, with JPM, provided feedback during the writing of the manuscript. JLL computed the TROPOMI-derived $NO_2$ lifetimes and provided advice on their use, and feedback on the manuscript.

*Competing interests.* We declare that we have no competing interests.

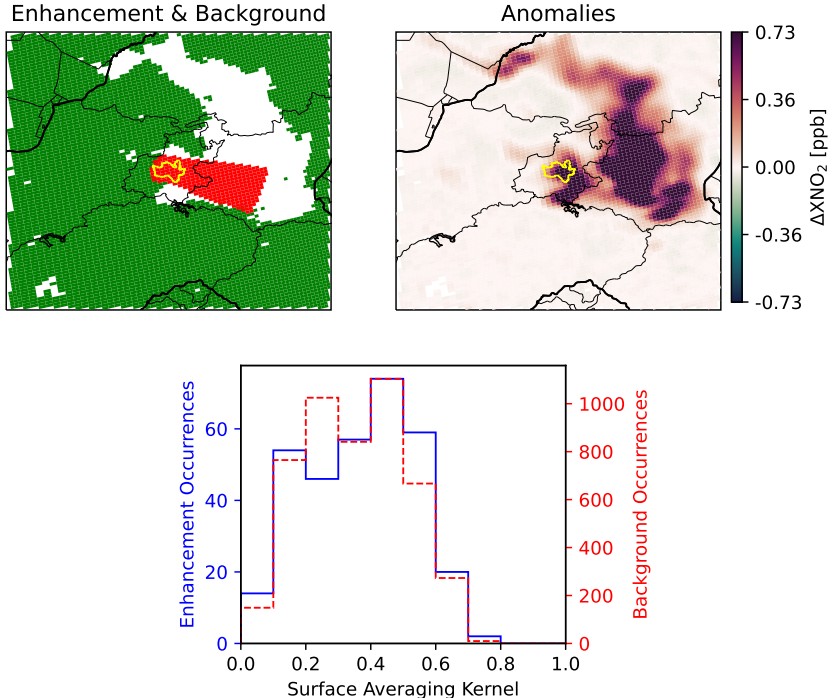

Johannesburg 10 May 2018

**Figure C2.** The lower plot shows the similarity of the distribution of TROPOMI $NO_2$ surface column averaging kernel values in the enhancement (solid blue histogram) and in the background (dashed red histogram). The geographical distribution of these surface column averaging kernels are shown in the top left map, where green points indicate the background region, and the red points indicate the enhancement region. The yellow solid line outlines the city. The top right map shows the $NO_2$ anomalies in the vicinity of the city.

*Acknowledgements.* This project is undertaken with the financial support of the Canadian Space Agency as part of the Earth System Science Data Analyses Program, Grant #16SUASCOBF. The authors gratefully acknowledge the efforts of the OCO-2, OCO-3, TROPOMI, EDGAR, ODIAC, FFDAS, MACCity, and Hestia teams for producing these valuable datasets. OCO-2 and OCO-3 data were produced by the OCO-2 and OCO-3 projects at the Jet Propulsion Laboratory, California Institute of Technology. A portion of this research was carried out at the Jet Propulsion Laboratory, California Institute of Technology, under a contract with the National Aeronautics and Space Administration (80NM0018D0004). The authors also wish to thank two anonymous reviewers for helpful and constructive comments that significantly strengthened the manuscript.

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
