# Peer review of "Estimating Enhancement Ratios of Nitrogen Dioxide, Carbon Monoxide and Carbon Dioxide using Satellite Observations"

_Atmospheric Chemistry and Physics, 2022_

## Author Comment (AC1)

**Response to Anonymous Referee #1**

We thank Referee #1 for helpful comments and suggestions. We have addressed each of the comments below. Referee comments are in *red italics* and our responses are in Roman font.

> *The methodology section would benefit greatly from the inclusion of more figures to illustrate the various steps of the analysis and aid the reader's understanding. I find this particularly important since several satellite products are used, each with distinct spatial coverage and overall data handling. For example, the differences in spatial coverage between the various satellite are not shown, nor is the colocation of OCO-2 and TROMPOMI/OCO-3 SAM data (section 3.1), or the masking that results from taking only using data from the sparser production (section 3.3.3). Similarly, the plume extraction presented in section 3.2 is not shown, nor is the impact of smoothing on the TROPOMI/OCO-3 data (section 3.3.1). At a minimum, a figure showing the different spatial sampling of the three satellites should be included.*

We have added an additional figure (below) to more clearly show the ground tracks of the three satellites we have used in the paper. These are examples of the various satellite instrument ground tracks over Buenos Aires taken on 11 January 2019 (top 4) and 19 September 2019 (bottom 2). Red points indicate the urban enhancement plume.

[Figure]

*There appear to be two distinct methods used for the enhancement calculation depending on which product is used in the analysis; a cross section taken downwind of the city (OCO) and the enhancement over the entire bounding area for the city (TROPOMI). The manuscript jumps around between these various methods, which is difficult to follow. I suggest clarifying the these two complementary methods within the text or with some sort of figure summarizing the methods.*

Yes, there are distinct methods as the referee has described depending on which ratio is being computed and from which instruments. We have clarified the text by adding the following sentences:

OCO-2 $CO_2$: "To compute enhancement ratios, coincident TROPOMI CO and $NO_2$ enhancements are selected at the locations of the OCO-2 ground track."

OCO-3 $CO_2$: "Coincident TROPOMI enhancements are selected at the locations of the OCO-3 SAM measurements."

$NO_2$:CO: "... only direct observation of the cities within their bounding areas are considered when deriving $NO_2$:CO enhancement ratios."

**Specific comments**

*Representativeness of enhancement ratios – Due to a limited number of overpasses (e.g. 1 for Toronto), how representative do you think some of these ratios are to the timescale of the inventories? I believe a brief discussion concerning this is warranted since the number of overpasses per city varies, while it appears conclusions about the validity of inventories are drawn from all cities regardless of the number of overpasses used in the ratio calculation.*

We have computed enhancement ratios for each overpass individually for each city to quantify the variability in enhancement ratios across overpasses. Most of the overpasses are in the summer, which would bias our results to summertime emissions, and, as you point out, in other cities, such as Toronto, the number of overpasses is extremely limited. As the satellite records lengthen, this representativeness bias should reduce somewhat. At the moment, however, most of the inventories are reported annually, so sub-sampling the inventories is not yet possible. We have added a brief discussion of the representativeness issue in the Discussion:

"The overpasses that successfully pass our filtering criteria are biased toward sunnier conditions and are most often collected in summertime, and some sites have very few overpasses (e.g., Toronto). If the enhancement ratios change seasonally, as expected, this type of analysis could cause a representativeness error, in which the comparisons between the measured enhancement ratios and the reported annual inventory ratios are systematically biased. Currently, the EDGAR and MACCity inventories, which provide CO and $NO_2$ emissions, do not report sub-annual emissions, so comparing to seasonal inventory ratios is not possible. With longer satellite time series providing more opportunities for wintertime enhancement ratios, we will be able to compute robust annual enhancement ratios to compare with the annual inventories."

*Do you expect the biosphere to impact your CO2 or CO enhancements, and subsequently your ratios?*

Yes, the biosphere will impact our enhancement ratios, but it should be a relatively small effect. Wu et al. [2022] calculate CO:$CO_2$ emission ratios over 4 cities including Los Angeles. In the paper they derive fossil fuel and biogenic anomalies using the X-STILT transport and SMuRF models. They find biogenic $XCO_2$ anomalies far smaller than those of fossil fuels. They provide the overpass example of Zibo on 21 June 2020, which contains the largest biogenic urban–background contrast that they studied. The biogenic anomalies ranged from 0 to 0.4 ppm while the sounding-level fossil fuel $CO_2$ enhancements ranged from 2 to 7 ppm. They find that typically the aggregated biogenic anomaly (i.e., summed for all footprints) stays low, with an absolute value of $< 0.3$ ppm. The aggregated fossil fuel $CO_2$ enhancements will be roughly 2 orders of magnitude larger (figure 2, S7, S11). We have added the following paragraph to the discussion:

"One potential source of error in this analysis is from biogenic emissions of CO and cycling of $CO_2$, but we expect these effects will be small. Under the assumption that there is less vegetation within urban boundaries than outside the urban region, biogenic CO and $CO_2$ emissions have the potential to affect the urban-rural

gradients, especially during the growing season. According to recent studies, however, these gradients are significantly smaller than the enhancements we measure, suggesting that urban CO and $CO_2$ enhancements are dominated by fossil fuel emissions [Plant et al., 2022; Wu et al., 2022]. Further modeling analysis will be necessary to apply this kind of analysis to smaller emission sources."

*Lines 191-194 – How are the secondary sources identified?*

The secondary sources from cities are identified using the European Commission Joint Research Centre's (EC-JRC) Global Human Settlement layer Urban Centres Database (GHS-UCDB). The secondary sources from power plants are identified using the Carbon Monitoring for Action (CARMA) database. We have added this to the text.

*Lines 194- 199 – What explains the discrepancy between the MERRA-2 winds and the plume direction, and the resulting variability in overpass retention rates? Is it errors in the wind direction and/or issues with the automatic filtering scheme since manual inspection and correction is required?*

The MERRA-2 model output is at a spatial resolution of 0.5° latitude × 0.625° longitude with a 3-hourly temporal resolution. We do not expect the wind direction to always be accurate at city scales and at the time of the overpass, and the automatic filtering simply ensures that the measurements are downwind of the source, and that the wind rotation is not too large. We have added the following text:

"Errors in wind direction can be caused by the inability of the coarse model resolution to resolve local topography, or if the 50-m winds are not representative of the winds at the local plume height. The wind rotation we perform should at least partially correct for both these errors."

*Figure 1 – How are the red points assigned to the city? Without more information or geographical details about the city and location of the overpass, it is hard to interpret.*

The new figure we have added shows the city and OCO-2 ground track, which we hope makes the assignment of the plume (red points) clearer. In what was previously Figure 1 (Figure 2 in the updated manuscript), we added clarifying text to the figure caption:

"Red points indicate the enhancement and are where the Gaussian plume intersects with the OCO-2 ground track."

*Line 273 – What is allowed to change in the bootstrap? Are the anomalies of each species within each overpass resampled?*

Within the bootstrap, *pairs* of anomalies are randomly selected and the slope is computed from those anomaly pairs. This selection process is repeated to compute the average slope and variability of the slope. This has been clarified in the paper in this sentence:

"Bootstrapping is a re-sampling technique in which random pairs of anomalies are drawn with replacement and fit independently, and has been used in previous enhancement ratio studies..."

*Lines 284-285 – "As we are correcting the enhancement ratios, the effect of dispersion cancels out in the ratio." This sentence is unclear to me. What dispersion?*

We were referring to the dispersion of the plume as it travels away from the city toward the OCO-2 ground track, but upon further reflection, this sentence is unnecessary in the manuscript and we have removed it.

*Figure 2 – The color scale in the 'Background' plot is hard to interpret using the color bar. I understand this the same color bar as the observed data which has a larger spread of values, but I cannot tell if it is all one value for the background or not. This might be an issue with my screen, but it is something to think about.*

We have used a different set of colour scales for the plots in this figure.

[Figure]

*Figure 4 & Section 3.4 – It was not until I saw Figure 4 that I realized that a single regression is calculated for all anomalies across multiple days. I would suggest explicitly stating this in the text since, in contrast, Figures 1 & 2 are for single days. In this analysis, are the number of anomaly data points that same for each overpass? If not, is there potential for a subset of overpasses with higher density of data (i.e. more points) to drive the combined regression?*

This is now explicitly stated in the text in §3.4. The number of points is not the same for each overpass, and this will generally bias the number of points to summer days or days that are less cloudy. We have added a brief discussion of representativeness error in the Discussion section.

"To determine enhancement ratios, we aggregate all overpasses for a given city and regress one set of anomalies onto the other using a reduced major axis regression..."

"The overpasses that successfully pass our filtering criteria are biased toward sunnier conditions, and are most often collected in summertime, and some sites have very few overpasses (e.g., Toronto). If the enhancement ratios change seasonally, as expected, this type of analysis could cause a representativeness error, in which the comparisons between the measured enhancement ratios and the reported annual inventory ratios are systematically biased. Currently, the EDGAR and MACCity inventories, which provide CO and $NO_2$ emissions, do not report sub-annual emissions, so comparing to seasonal inventory ratios is not possible. With longer satellite time series providing more opportunities for wintertime enhancement ratios, we will be able to compute robust annual enhancement ratios to compare with the annual inventories."

*Figure 5-7 – The number of overpasses that each measurement ratio is based upon should be included in the figure. It seems that this information is at least partially available in the appendix, so another option would be to include a reference to those tables in the caption.*

We have update Tables 2 & 3 to include the number of overpasses used in each ratio.

**Table 2.** All enhancement ratios derived using OCO-2/3 and TROPOMI. Overpasses must individually have sufficient linear dependence (R> 0.2). Cities without any such overpasses are marked by a dash.

| City | No. of OCO-2 / 3 Overpasses | CO:CO2 Enhancement Ratio [ppb:ppm] | No. of OCO-2 / 3 Overpasses | NO2:CO2 Enhancement Ratio [ppb:ppm] | No. of TROPOMI Overpasses | NO2:CO Enhancement Ratio [ppb:ppb] |
|---|---|---|---|---|---|---|
| Ahmadabad | 3 | $12.7 \pm 1.1$ | - | - | - | - |
| Alexandria | - | - | 5 | $0.39 \pm 0.04$ | 30 | $0.018 \pm 0.002$ |
| Baghdad | 2 / 2 | $6.3 \pm 0.3$ | 6 / 2 | $0.89 \pm 0.04$ | 67 | $0.176 \pm 0.013$ |
| Buenos Aires | 6 / 4 | $7.9 \pm 0.3$ | 6 / 4 | $0.57 \pm 0.02$ | 39 | $0.048 \pm 0.004$ |
| Cairo | 12 / 2 | $14.0 \pm 0.9$ | 12 / 2 | $0.49 \pm 0.02$ | 84 | $0.037 \pm 0.001$ |
| Chicago | 5 | $13.3 \pm 0.7$ | 5 | $0.61 \pm 0.03$ | 44 | $0.038 \pm 0.005$ |
| Dallas | 3 / 1 | $3.7 \pm 0.2$ | 3 / 1 | $0.18 \pm 0.01$ | 57 | $0.032 \pm 0.002$ |
| Delhi | 3 / 1 | $13.0 \pm 0.5$ | 4 / 1 | $0.79 \pm 0.04$ | 46 | $0.017 \pm 0.001$ |
| Guangzhou | 3 | $13.2 \pm 1.6$ | - | - | 23 | $0.014 \pm 0.002$ |
| Houston | - | - | 5 / 1 | $0.36 \pm 0.02$ | 54 | $0.015 \pm 0.001$ |
| Indianapolis | 7 | $13.2 \pm 0.9$ | 7 | $0.41 \pm 0.02$ | 32 | $0.018 \pm 0.003$ |
| Johannesburg | 11 | $15.4 \pm 0.5$ | 11 | $2.54 \pm 0.16$ | 62 | $0.180 \pm 0.012$ |
| Lahore | - | - | - | - | 42 | $0.021 \pm 0.002$ |
| Las Vegas | 10 / 1 | $8.1 \pm 0.5$ | 10 / 1 | $0.41 \pm 0.02$ | 58 | $0.021 \pm 0.001$ |
| London | 1 / 1 | $5.7 \pm 0.3$ | 1 / 1 | $0.50 \pm 0.01$ | 47 | $0.071 \pm 0.006$ |
| Los Angeles | 7 / 2 | $8.2 \pm 0.3$ | 7 / 2 | $0.32 \pm 0.01$ | 84 | $0.031 \pm 0.001$ |
| Madrid | - | - | 9 / 1 | $0.92 \pm 0.04$ | 49 | $0.097 \pm 0.007$ |
| Mexico City | 7 / 5 | $19.1 \pm 0.7$ | 7 / 5 | $1.24 \pm 0.05$ | 50 | $0.041 \pm 0.003$ |
| Moscow | 5 | $12.5 \pm 0.7$ | 5 | $1.13 \pm 0.06$ | 53 | $0.082 \pm 0.007$ |
| Paris | 4 / 2 | $12.4 \pm 1.0$ | 4 / 2 | $0.70 \pm 0.04$ | 39 | $0.052 \pm 0.007$ |
| Phoenix | 14 / 6 | $6.6 \pm 0.2$ | 14 / 6 | $0.30 \pm 0.01$ | 74 | $0.028 \pm 0.001$ |
| Riyadh | 8 | $11.2 \pm 0.5$ | - | - | 52 | $0.359 \pm 0.030$ |
| San Francisco | - | - | 7 | $0.18 \pm 0.01$ | 62 | $0.009 \pm 0.001$ |
| Seoul | - | - | 4 | $0.698 \pm 0.04$ | 45 | $0.066 \pm 0.008$ |
| Tehran | 10 / 3 | $23.9 \pm 1.2$ | 10 / 3 | $1.78 \pm 0.16$ | 86 | $0.051 \pm 0.001$ |
| Toronto | 1 | $10.6 \pm 1.2$ | 1 | $0.80 \pm 0.07$ | 39 | $0.026 \pm 0.005$ |
| Vienna | 2 | $7.2 \pm 0.6$ | - | - | - | - |

**Table 3.** All NO2:CO2 and NO2:CO enhancement ratios derived using OCO-2/3 and TROPOMI, including NO2 lifetime correction. Overpasses must individually have sufficient linear dependence (R> 0.2). Cities without any such overpasses are marked by a dash.

| City | No. of OCO-2 / 3 Overpasses | Average Wind Speed for Correction (m/s) | NO2:CO2 Enhancement Ratio [ppb:ppm] | No. of TROPOMI Overpasses | Average Wind Speed for Correction (m/s) | NO2:CO Enhancement Ratio [ppb:ppb] |
|---|---|---|---|---|---|---|
| Alexandria | 5 | 5.13 | $0.44 \pm 0.10$ | 30 | 6.30 | $0.019 \pm 0.005$ |
| Baghdad | 6 / 2 | 5.79 | $1.12 \pm 0.23$ | 67 | 6.91 | $0.210 \pm 0.044$ |
| Buenos Aires | 6 / 4 | 4.33 | $1.26 \pm 0.25$ | 39 | 6.58 | $0.074 \pm 0.015$ |
| Cairo | 12 / 2 | 4.39 | $0.85 \pm 0.17$ | 84 | 5.02 | $0.092 \pm 0.018$ |
| Chicago | 5 | 4.49 | $1.20 \pm 0.24$ | 44 | 4.75 | $0.064 \pm 0.014$ |
| Dallas | 3 / 1 | 4.41 | $0.33 \pm 0.07$ | 57 | 4.37 | $0.119 \pm 0.024$ |
| Delhi | 4 / 1 | 2.15 | $5.83 \pm 1.17$ | 46 | 4.65 | $0.047 \pm 0.010$ |
| Guangzhou | - | - | - | 23 | 3.88 | $0.286 \pm 0.057$ |
| Houston | 5 / 1 | 4.94 | $0.74 \pm 0.15$ | 56 | 3.84 | $0.058 \pm 0.012$ |
| Indianapolis | 7 | 4.45 | $0.55 \pm 0.11$ | 32 | 4.29 | $0.032 \pm 0.007$ |
| Johannesburg | 11 | 5.20 | $2.77 \pm 0.58$ | 62 | 5.55 | $0.193 \pm 0.041$ |
| Lahore | - | - | - | 42 | 2.73 | $0.072 \pm 0.014$ |
| Las Vegas | 10 / 1 | 4.44 | $0.89 \pm 0.18$ | 58 | 3.87 | $0.052 \pm 0.011$ |
| London | 1 / 1 | 6.86 | $0.63 \pm 0.13$ | 47 | 4.38 | $0.118 \pm 0.024$ |
| Los Angeles | 7 / 2 | 3.59 | $0.68 \pm 0.14$ | 84 | 4.34 | $0.058 \pm 0.012$ |
| Madrid | 9 / 1 | 5.22 | $1.71 \pm 0.34$ | 49 | 3.92 | $0.208 \pm 0.042$ |
| Mexico City | 7 / 5 | 2.80 | $3.79 \pm 0.76$ | 50 | 2.56 | $0.145 \pm 0.029$ |
| Moscow | 5 | 5.92 | $2.08 \pm 0.42$ | 53 | 5.42 | $0.159 \pm 0.033$ |
| Paris | 4 / 2 | 5.44 | $1.29 \pm 0.26$ | 39 | 3.51 | $0.188 \pm 0.038$ |
| Phoenix | 14 / 6 | 4.17 | $2.01 \pm 0.41$ | 74 | 3.59 | $0.386 \pm 0.078$ |
| Riyadh | - | - | - | 52 | 7.30 | $0.505 \pm 0.105$ |
| San Francisco | 7 | 4.73 | $0.22 \pm 0.05$ | 62 | 4.80 | $0.010 \pm 0.002$ |
| Seoul | 4 | 7.57 | $1.42 \pm 0.29$ | 45 | 3.55 | $0.303 \pm 0.061$ |
| Tehran | 10 / 3 | 3.26 | $3.89 \pm 0.79$ | 86 | 5.28 | $0.079 \pm 0.016$ |
| Toronto | 1 | 2.88 | $2.47 \pm 0.50$ | 39 | 4.29 | $0.056 \pm 0.012$ |

*Figure 5 – I cannot see where the "light blue areas" are on top of the measured ratios. I see black error bars, so perhaps this is a typo.*

This is indeed a typo that refers to an older version of the figure. It has been fixed in the manuscript.

*Lines 355-358 – Is the comparison for cities both in this work and Lama et al. shown anywhere, in something like a figure or table? If not, I believe this should be added, possibly to the appendix. If this comparison is buried into one of the figures, a reference to it should be added to the text.*

We have added the following figure to the paper to show the comparison between Lama et al. (2020) and our work. We have also modified our discussion in section 4.2. Our previous discussion was with reference to incorrect values in the appendix of Lama et al. (2020). The authors have been contacted and are in communication with the journal to fix this issue. We include their corrected numbers in our paper.

With the corrected numbers from Lama et al., our results show good agreement, within the uncertainties, of corrected NO2:CO enhancement ratios at all cities except Tehran. This has also been added to our discussion.

[Figure]

*In the introduction, you mention the TIMES scaling of CO2 by Nassar et al. Do you apply those scale factors to account for the diurnal variability of CO2, since your measurements are based on afternoon overpasses? You mention this correction (the magnitude of which is not included) in section 5.2 for LA and Indianapolis, however, it is not clear if these corrections are applied to the ratios shown in Figures 5 &6.*

Yes, the TIMES scaling is added to the ratios computed from the inventories for the reasons stated above. We have added clarification in the text in §2.5.

*Lines 425-429 – You present the NO2 emission estimates based on ratios with and without the NO2 lifetime correction. The reader should not assume the un-corrected emission rate is accurate, right? If this is the case, I would suggest you state that because as written it is not clear which value for NO2 is the one the reader should remember.*

That is correct. We have rephrased the sentence:

"... we estimate emissions of $NO_2$ within the SoCAB to be $89 \pm 17$ $GgNO_2/yr$ (after the $NO_2$ lifetime correction is applied), which agrees with the CARB estimate for 2015 (105 $GgNO_2/yr$). However, it is smaller than the annual EDGAR estimate of 132 $GgNO_2/yr$, and larger than the MACCity estimate of 43.3 $GgNO_2/yr$..."

*Line 455 – underestimations of which species?*

We are referring to CO. This has been clarified in the text.

*Table 2 – What metric is used to discriminate a "poor" linear relationship? Low correlation coefficient, low R?*

We use $R < 0.2$ as the metric. We've added this to the Table 2 & 3 captions.

*Tables 2, 3 – You should include how many TROPOMI overpasses are used to generate the NO2:CO enhancement ratios for each city.*

These values have been added to tables 2 & 3.

**References**

Wu, D., Liu, J., Wennberg, P. O., Palmer, P. I., Nelson, R. R., Kiel, M., and Eldering, A.: Towards sector-based attribution using intra-city variations in satellite-based emission ratios between CO2 and CO, Atmos. Chem. Phys., 22, 14547–14570, https://doi.org/10.5194/acp-22-14547-2022, 2022.

Plant, G., Kort, E. A., Murray, L. T., Maasakkers, J. D., Aben, I.: Evaluating urban methane emissions from space using TROPOMI methane and carbon monoxide observations, Remote Sensing of Environment, Volume 268, 2022, 112756, ISSN 0034-4257, https://doi.org/10.1016/j.rse.2021.112756.

---

## Author Comment (AC2)

**Response to Anonymous Referee #2**

We thank Referee #2 for thoughtful comments and suggestions. We have addressed each of the comments below. Referee comments are in *red italics* and our responses are in Roman font.

> *In order to derive the emission ratios, using total column dry air-column mole fractions from satellite sounders one needs the know to what extent the observed variations in clean vs. urban air emissions are translated into the satellite signal. For NO2:CO ratios this was done by Lama et al. (2020) using CAMS simulated profiles onto which the total column averaging kernel was applied. Here the anomalies are divided by the surface averaging kernel values (I assume it corresponds with the value in the total column averaging kernel array that corresponds to the lowest altitude layer although this is not that clearly stated). Using these column averaging kernel values to look into aspects that pertain to a specific partial column, comes with its own set of uncertainties that need to be discussed. For instance the column averaging kernel values correspond to specific airmass layers with specific dimensions. These vertical dimensions could differ between sounders (and retrieved species), the Planetary Boundary Layer under consideration etc., and with that, biases could be induced into the analysis. This is particularly relevant for NO2, which features relatively low kernel values near the surface, but also CO which features an equally strong gradient near the surface (an additional figure showing typical total column averaging kernel profile shapes would be a useful addition to the paper).*

We use the column averaging kernel value at the surface pressure value of the measurement. This avoids issues with different retrieval grids and layers. We have now stated this explicitly. We have also included an example of the averaging kernel profiles at an enhancement in Appendix C.

> *Related to this, dividing the anomaly by the surface averaging kernel will only yield correct results if there is no a priori contribution to the anomaly (i.e. the a priori used in the satellite retrieval may not differ between what is considered background and urban air). This needs to be verified and clearly stated.*

Thank you for pointing this out. The a priori $CO_2$ profiles for OCO-2 and OCO-3 depend only on dynamical latitude and time, and have no knowledge of urban vs rural locations. Therefore, the current method of dividing the measured anomaly by the surface pressure averaging kernel is valid for our $CO_2$ anomalies.

However, TROPOMI uses priors extracted from the TM5 chemical transport model on a 1° x 1° grid, and therefore the priors will contain urban enhancements. This requires a reformulation of the averaging kernel weighting, requiring a correction term:

$$\Delta c^t = \frac{\Delta \hat{c}}{a_0} - \frac{(1 - a_0)\Delta c^a}{a_0}, \tag{1}$$

where $\Delta c^t$ is the true anomaly, $\Delta \hat{c}$ is the measured anomaly, $\Delta c^a$ is the prior anomaly ($\Delta c^a = c_u^a - c_b^a$, where $c_u^a$ and $c_b^a$ are the urban and background a priori columns, respectively), and $a_0$ is the surface pressure value of the column averaging kernel. The second term on the right hand side of the equation is required when $c_u^a - c_b^a \neq 0$, and reduces in magnitude as $a_0$ approaches 1. Equation 1, however, is only valid if the surface pressure averaging kernel value inside and outside the urban plume is the same (or similar enough). If the averaging kernel is tightly correlated with the atmospheric column, this equation is no longer correct, in general. The figure below provides an example of the similar distribution of surface averaging kernel values in the enhancement and background. The TROPOMI averaging kernels appear to be relatively insensitive to the plume itself, so we believe that Equation 1 is valid for our purposes. We have added the figure to the manuscript.

In the paper, we included the second term in our calculations and have rerun the analyses. The correction term was small for both $NO_2$ and CO, so the main results of the paper are unchanged, but the anomalies and enhancement ratios all changed slightly. We've also added an appendix to the paper containing the full derivation of Equation 1.

[Figure]

That said, we wanted to evaluate the magnitude of the error caused by excluding the extra correction term, so we performed a sensitivity analysis to quantify the magnitude of the bias caused by using the first term in Equation 1 over a range of surface column averaging kernel values and a range of a priori enhancements. We have not added this analysis to the paper, but it is here for completeness. In the analysis that follows, all the data are synthetic; there are no atmospheric measurements involved.

Figures 1–5 show in the top panel the true (modeled) enhancement in blue with square markers, the correction term (second term on the right hand side of Equation 1) in red with "+" markers, the approximation to the true enhancement (first term on the right hand side of Equation 1) in yellow lines with dots, and the simulated "measured" enhancement ($\Delta\hat{c}$) in purple lines with circles. All are plotted as a function of the surface pressure column averaging kernel value. The bottom panel shows the percent difference between the approximation to the true enhancement and the true enhancement (yellow), and the measured enhancement to the true enhancement (purple). Figure 1 shows that when the priors do not have an enhancement, the first term on the right hand side of Equation 1 is exact. Figure 2 shows the results when the a priori enhancement is equal to the measured enhancement. In this case, it would be more accurate to report the measured enhancement without a correction at all. The truth likely lies somewhere between these cases, where the retrieval makes a small adjustment to the a priori enhancement. If the a priori enhancement is between 0.5 and 1.5 times the measured enhancement, using the uncorrected measured enhancements incurs less error than using the first term on the right hand side of equation 1 alone (Figs. 3, 4). If the a priori enhancement is less than 0.5 of the measured enhancement, it is better to use the first term on the right hand side of equation 1 than the uncorrected enhancement (Fig. 5).

[Figure]

Figure 1: The impact of omitting the additional correction term (second term on the right hand side of Equation 1) on the corrected enhancement. In this case, the prior enhancement is set to 0, and therefore the first term on the right hand side of Equation 1 exactly reproduces the true enhancement.

[Figure]

Figure 2: The impact of omitting the additional correction term (second term on the right hand side of Equation 1) on the corrected enhancement. In this case, the prior enhancement is set to be equal to the measured enhancement, and therefore the measured results are independent of the surface pressure averaging kernel value, and a better approximation to the truth than the first term on the right hand side of Equation 1.

[Figure]

Figure 3: The impact of omitting the additional correction term (second term on the right hand side of Equation 1) on the corrected enhancement. In this case, the prior enhancement is set to be 33% larger than the measured enhancement. In this case, incorrectly adjusting the measured enhancement results in larger biases than using an uncorrected measured enhancement.

[Figure]

Figure 4: The impact of omitting the additional correction term (second term on the right hand side of Equation 1) on the corrected enhancement. In this case, the prior enhancement is set to be 33% smaller than the measured enhancement. In this case, incorrectly adjusting the measured enhancement results in larger biases than using an uncorrected measured enhancement.

[Figure]

Figure 5: The impact of omitting the additional correction term (second term on the right hand side of Equation 1) on the corrected enhancement. In this case, the prior enhancement is set to be 83% smaller than the measured enhancement. In this case, incorrectly adjusting the measured enhancement results in smaller biases than using an uncorrected measured enhancement.

*Secondly, some of the obtained emission ratios are derived from a very limited set of data. Foremost among these is Toronto with only one OCO-2/3 overpass. This begs the question to what extent this affects the overall uncertainty. Valid indicative information in this regard would be to perform the analysis for better sampled stations on an overpass per overpass basis and see what the obtained range of results would be and whether or not there is a temporal/seasonal aspect to the variability.*

We have performed the suggested analysis and include a figure below, showing that the overpasses are generally biased to summertime, but that we do not see much systematic seasonality in the enhancement ratios. That said, this representativeness error is not quantified in our analysis, and we have included a paragraph discussing this effect.

[Figure]

"A second possible source of error is the temporal representativeness of the satellite data used in this analysis. The overpasses that successfully pass our filtering criteria are biased toward sunnier conditions, and are most often collected in summertime, and some sites have very few overpasses (e.g., Toronto). If the enhancement ratios change seasonally, as expected, this type of analysis could cause a representativeness error, in which the comparisons between the measured enhancement ratios and the reported annual inventory ratios are systematically biased. Currently, the EDGAR and MACCity inventories, which provide CO and $NO_2$ emissions, do not report sub-annual emissions, so comparing to seasonal inventory ratios is not possible. With longer satellite time series providing more opportunities for wintertime enhancement ratios, we will be able to compute robust annual enhancement ratios to compare with the annual inventories."

**Specific comments**

*Line 103: add references for each of these products*

Done.

*Line 188: concerning 'points away': Is there a degree threshold?*

There is no explicit maximum angle threshold in our analysis, but the maximum angle between the wind and the center of the enhancement we use in this study is 66°; typically, the angle is less than 45°. In our analysis, the angle is permitted to be larger when the OCO-2 ground track is closer to the city. We have clarified in the text:

"When the boundary layer wind direction does not intersect the OCO-2 ground track the overpass is rejected, as the pollution plume from the city will not be captured."

*Line 194: how are these corrections implemented? Could the misdirection of the plume also be related to the injection height of the emissions?*

It is possible that if the plumes are injected significantly above 50 m, (or if vertical mixing is much more vigorous than we expect), the 50 m winds we use would be inaccurate or inappropriate. We have added text to make it clear in the paper that we have assumed that plumes are injected around or below 50 m and that if this assumption is not true, the wind correction could partially correct for that:

"Errors in wind direction can be caused by the inability of the coarse model resolution to resolve local topography, or if the 50-m winds are not representative of the winds at the local plume height. The wind rotation we perform should at least partially correct for both these errors."

*Line 228: Do you use certain fixed criteria to do this manual selection?*

We follow the manual selection methods used in Nassar et al. [2017, 2021], and visually identify a clear drop in $XCO_2$ to mark the end of the plume. An automated approach is worth considering for future work, but we have not yet identified a robust metric.

"a maximum downwind distance for the plume is determined manually, following Nassar et al. [2017;2021], to visually identify a drop in $XCO_2$, which limits the length of the plume to an area where significant enhancements are observed."

*Line 291: assuming equal area?*

Correct. This has been clarified in the text.

*Line 295: 20% of the uncertainty itself, the initial emission ratio, the final emission ratio, the correction?*

We add a flat 20% in quadrature to the initial enhancement ratio uncertainty. This has been clarified in the text.

*Line 357-364: It is stated that the differences between this study and Lama et al. prior to the lifetime correction falls within 25% for most sites. Could you discuss these differences in light of your (and Lama et al.) uncertainty estimates? An additional figure would also help the discussion. The authors also point to the difference in sampling rate (each overpass vs. a limited number of*

*days in Lama et al.) but this aspect alone seems inconclusive to describe the particularly large discrepancies at some sites.*

We have updated our comparisons with the Lama et al. paper. Our previous discussion was with reference to incorrect values in the appendix of Lama et al. (2020). We contacted the authors and they are in communication with the journal to fix this issue. We also added a figure to the text to show the differences more clearly. Our results show good agreement with the Lama results, within the uncertainties, of lifetime corrected $NO_2$:CO enhancement ratios at all cities except Tehran. An additional discussion has been added to the text.

[Figure]

*Line 448: Are we referring to lifetime corrected or uncorrected data? This is sometimes unclear in this section of the document. The agreement is sometimes different by several factors. 'Good' might therefore be nuanced.*

We are focusing on the lifetime corrected results in this section, and we have added to the text to make this clear. The additional figure (see above) should help the reader understand what we mean by "good" agreement with the Lama et al. results.

*Figure 1: It is unclear to me what the red area points to. It seems that this would be the downwind area that is considered for the analysis. But it also features some upwind areas?*

In Figure 1 of the original manuscript, the data are along the OCO-2 ground track, and the OCO-2 track is downwind of the city. The red points are where the Gaussian plume intersects the OCO-2 track. We have clarified the caption to describe this more fully. We have also added a new figure (below) that that depicts the various satellite ground tracks that we hope will clarify where the red points come from.

[Figure]

*Figures 6: Particularly Delhi features a correction that is larger than a factor 6. Could you elaborate why this particular city stands out?*

We believe this is caused by particularly low wind conditions. The wind speeds used for the correction have been added to Table 3. We have added a comment about this in the text.

"Delhi has a particularly low wind speed (Table 3), which may cause the $NO_2$ lifetime correction to be overestimated."

**References**

Martínez-Alonso, S., Deeter, M., Worden, H., Borsdorff, T., Aben, I., Commane, R., Daube, B., Francis, G., George, M., Landgraf, J., Mao, D., McKain, K., and Wofsy, S.: 1.5 years of TROPOMI CO measurements: comparisons to MOPITT and ATom, Atmos. Meas. Tech., 13, 4841–4864, https://doi.org/10.5194/amt-13-4841-2020, 2020.

Borsdorff, T., aan de Brugh, J., Hu, H., Hasekamp, O., Sussmann, R., Rettinger, M., Hase, F., Gross, J., Schneider, M., Garcia, O., Stremme, W., Grutter, M., Feist, D. G., Arnold, S. G., De Mazière, M., Kumar Sha, M., Pollard, D. F., Kiel, M., Roehl, C., Wennberg, P. O., Toon, G. C., and Landgraf, J.: Mapping carbon monoxide pollution from space down to city scales with daily global coverage, Atmos. Meas. Tech., 11, 5507–5518, https://doi.org/10.5194/amt-11-5507-2018, 2018.

Laughner, J. L., Roche, S., Kiel, M., Toon, G. C., Wunch, D., Baier, B. C., Biraud, S., Chen, H., Kivi, R., Laemmel, T., McKain, K., Quéhé, P.-Y., Rousogenous, C., Stephens, B. B., Walker, K., and Wennberg, P. O.: A new algorithm to generate a priori trace gas profiles for the GGG2020 retrieval algorithm, Atmos. Meas. Tech. Discuss. [preprint], https://doi.org/10.5194/amt-2022-267, in review, 2022.